# Dynamic Novel View Synthesis in High Dynamic Range

Kaixuan Zhang[1]    Zhipeng Xiong[1]    Minxian Li[1,2]*    Mingwu Ren[1,2]

Jiankang Deng[3]    Xiatian Zhu[4]

[1]Nanjing University of Science and Technology
[2]State Key Laboratory of Intelligent Manufacturing of Advanced Construction Machinery
[3]Imperial College London
[4]University of Surrey

## Abstract

High Dynamic Range Novel View Synthesis (HDR NVS) seeks to learn an HDR 3D model from Low Dynamic Range (LDR) training images captured under conventional imaging conditions. Current methods primarily focus on static scenes, implicitly assuming all scene elements remain stationary and non-living. However, real-world scenarios frequently feature dynamic elements, such as moving objects, varying lighting conditions, and other temporal events, thereby presenting a significantly more challenging scenario. To address this gap, we propose a more realistic problem named HDR Dynamic Novel View Synthesis (HDR DNVS), where the additional dimension "Dynamic" emphasizes the necessity of jointly modeling temporal radiance variations alongside sophisticated 3D translation between LDR and HDR. To tackle this complex, intertwined challenge, we introduce HDR-4DGS, a Gaussian Splatting-based architecture featured with an innovative dynamic tone-mapping module that explicitly connects HDR and LDR domains, maintaining temporal radiance coherence by dynamically adapting tone-mapping functions according to the evolving radiance distributions across the temporal dimension. As a result, HDR-4DGS achieves both temporal radiance consistency and spatially accurate color translation, enabling photorealistic HDR renderings from arbitrary viewpoints and time instances. Extensive experiments demonstrate that HDR-4DGS surpasses existing state-of-the-art methods in both quantitative performance and visual fidelity. Source code is available at `https://github.com/prinasi/HDR-4DGS`.

## 1 Introduction

Recent years have seen remarkable progress in Novel View Synthesis (NVS), which reconstructs 3D scene representations from multi-view images to enable photorealistic rendering from arbitrary viewpoints. These advances have fueled applications in gaming (Niedermayr et al., 2024), AR / VR (Zhou et al., 2018), and autonomous driving (Wang et al., 2024b). However, most existing NVS methods (Gao et al., 2022; Wu et al., 2024c) operate under two limiting assumptions: static scenes and low dynamic range inputs. These constraints significantly hinder their applicability in real-world environments, which often exhibit complex motion, time-varying illumination, and sensor-imposed luminance clipping.

Dynamic Novel View Synthesis (DNVS) extends NVS to scenes with temporal dynamics, reconstructing 4D radiance fields coherent in both space and time. Recent efforts in DNVS (Fang et al., 2022; Gao et al., 2021; Tian et al., 2023; Duan et al., 2024; Yang et al., 2024; Cho et al., 2024) have made notable progress in modeling dynamic geometry and appearance. However, they remain restricted to LDR imagery, which fails to capture the full spectrum of scene radiance. As a result, these methods struggle under high-contrast conditions (*e.g.*, direct sunlight or low-light environments) where overexposure and underexposure lead to significant information loss. More

---

*Minxian Li (minxianli@njust.edu.cn) is the corresponding author with School of Computer Science and Engineering, Nanjing University of Science and Technology.

fundamentally, LDR imaging is inherently misaligned with human visual perception, even under moderate lighting conditions. While the human eye can adapt to luminance levels spanning over ten orders of magnitude (Reinhard, 2020), typical LDR sensors cover only a narrow dynamic range. In addition, LDR images compress radiance through nonlinear camera response functions (CRFs), distorting local contrast, suppressing brightness gradients, and reducing color fidelity. These limitations not only impair perceptual quality under extreme lighting but also degrade the realism of ordinary scenes.

High Dynamic Range (HDR) imaging addresses these shortcomings by capturing a significantly broader range of luminance and color. By preserving detail in both highlights and shadows and maintaining fine-grained contrast, HDR techniques provide a closer match to the perceptual capabilities of the human visual system. Recent works in HDR NVS (Huang et al., 2022; Cai et al., 2024; Liu et al., 2025a) have attempted to reconstruct HDR content from multi-exposure LDR images of calibrated sensors. However, these methods are restricted to static scenes, implicitly assuming that all elements remain fixed over time. In contrast, real-world HDR scenarios are inherently dynamic, often involving moving objects, shifting illumination, and transient phenomena. These dynamics violate the assumptions of existing HDR NVS pipelines and introduce significant challenges: non-rigid motion and temporal variation create complex spatiotemporal inconsistencies, while the lack of reliable luminance priors from sparse LDR observations leads to severe photometric ambiguities.

To bridge this gap, we introduce High Dynamic Range Dynamic Novel View Synthesis (*HDR DNVS*), a new, more practical task that seeks to reconstruct temporally coherent HDR radiance fields and dynamic geometry from sparse, time-varying LDR inputs. HDR DNVS demands the joint modeling of evolving scene structure and HDR radiance, posing both geometric and photometric challenges absent in prior static or LDR-constrained settings. To tackle this, we propose ***HDR-4DGS***, a novel framework based on Gaussian Splatting (Kerbl et al., 2023) and equipped with a biologically inspired dynamic tone-mapping module. Drawing inspiration from human visual adaptation (Clifford et al., 2007), where retinal photoreceptors dynamically adjust to ambient brightness, HDR-4DGS includes a dynamic radiance context learner that models temporal radiance distributions. This is followed by per-channel tone-mapping functions that connect LDR representations with HDR space in a temporally adaptive and spatially accurate manner. This module explicitly bridges the LDR-HDR gap while maintaining radiance consistency and chromatic fidelity across time and space. Although conceptually straightforward, our proposed model embodies an intuitively elegant and computationally efficient design, which strategically adapts established sequential modeling techniques to address the core challenges of HDR DNVS with precision.

Our **contributions** are: (I) We introduce the HDR DNVS problem for the first time, which requires learning 4D HDR radiance fields with dynamic geometry and temporally coherent appearance from sparse LDR input. (II) We propose HDR-4DGS, a novel Gaussian Splatting architecture featuring dynamic tone-mapping for adaptively bridging the LDR and HDR domains under complex spatiotemporal variations. (III) To enable quantitative evaluation of HDR DNVS methods, we introduce HDR-4D-Syn and HDR-4D-Real — two novel benchmark datasets comprising 8 high-fidelity synthetic scenes and 4 real-world captured sequences, respectively. Each scene is meticulously annotated with ground-truth HDR images, time-varying 3D geometry, and synchronized multi-view LDR observations. (IV) Extensive experiments show that HDR-4DGS achieves state-of-the-art performance in both quantitative metrics and perceptual quality on challenging dynamic scenes.

## 2   RELATED WORK

**Novel view synthesis (NVS)** has seen transformative progress via neural rendering techniques. Early multi-view geometric methods (Schonberger & Frahm, 2016; Wang et al., 2021) face limitations in handling occlusions, textureless regions, and computing efficiency (Jiang, 2023). Modern approaches leverage continuous scene representations via deep networks, such as Neural Radiance Fields (NeRF) (Mildenhall et al., 2021) and the variants (Deng et al., 2022; Roessle et al., 2022; Wei et al., 2021; Xu et al., 2022) establishing coordinate-space implicit modeling for photorealistic synthesis under spatial smoothness constraints. However, NeRF's ray-marching paradigm suffers from high computational costs (Luo et al., 2024). In contrast, 3DGS (Kerbl et al., 2023) introduces an efficient point-based representation by parameterizing scenes as anisotropic 3D Gaussians, decoupling geometry and appearance while enabling real-time rendering with complex visual effects (*e.g.*,

specular highlights). Recent advances enhance 3DGS through frequency-domain supervision (Liang et al., 2024), depth-regularized optimization (Chung et al., 2024; Kung et al., 2024; Li et al., 2024), and memory-efficient designs (Wang et al., 2024c; Chen et al., 2024; Fan et al., 2024; Lu et al., 2024; Yang et al., 2025), achieving superior rendering quality. Nevertheless, these methods predominantly focus on static LDR scenes, limiting their applicability to real-world dynamic scenarios.

**HDR novel view synthesis (HDR NVS)** aims to reconstruct HDR scenes from multi-view LDR images. Early works like HDR-NeRF (Huang et al., 2022) extend neural radiance fields by incorporating an MLP-based tone-mapping module to bridge physical radiance and digital color spaces. To realize real-time rendering, HDR-GS (Cai et al., 2024) introduces a 3DGS framework with a neural tone-mapper that explicitly models HDR-to-LDR radiance transformations, achieving real-time HDR rendering while surpassing NeRF-based quality. Recently, GaussHDR (Liu et al., 2025a) proposes to unify 3D and 2D tone mapping in 3D Gaussian Splatting to facilitate HDR rendering. However, a critical limitation of these methods is their exclusive focus on *static scenes*; none explicitly model temporal dynamics, both in spatial and color space. Consequently, they are fundamentally unable to address the core challenges posed by real-world scenarios involving time-varying geometry, non-rigid motion, or temporally evolving illumination.

Indeed, Wu et al. (2024a) preliminarily investigated dynamic HDR reconstruction from LDR sequences. Interestingly, its has never carefully evaluated the HDR output, nor verifies its performance on real-world scenes, leaving the HDR DNVS problem largely under-explored. To address these issues, we introduce a purposed benchmark by carefully re-engineering its dataset and capture a real-world dataset additionally, including LDR/HDR paired training imagery and HDR quantitative assessment. We further propose a novel HDR DNVS model, HDR-4DGS, which delivers significantly more accurate HDR renderings while operating an order of magnitude faster.

**Dynamic novel view synthesis (DNVS)** focuses on modeling dynamic scenes with time-varying geometry and radiance. The key challenge lies in capturing intrinsic spatiotemporal correlations. Building upon NeRF (Mildenhall et al., 2021), two primary approaches have emerged: *i*) implicit / explicit spatiotemporal representations decompose scenes into time-aware feature grids to learn 6D plenoptic functions (Cao & Johnson, 2023; Li et al., 2022b; Fridovich-Keil et al., 2023; Wang et al., 2023), and *ii*) deformation-aware fields model motion through deformable transformations (Pumarola et al., 2021; Song et al., 2023; Abou-Chakra et al., 2024). Recent advances leverage 3DGS for dynamic rendering along two directions: *i*) deformation-based models (Wu et al., 2024b; Kratimenos et al., 2024; Bae et al., 2024; Shan et al., 2025) maintain canonical Gaussians deformed via time-varying fields, trading precise motion tracking for temporal continuity; *ii*) hyper-dimensional representations (Yang et al., 2024; Duan et al., 2024) extend Gaussians to 4D by introducing temporal centers and spatiotemporal rotations. However, all these methods are limited to LDR outputs, and we extend this to high-fidelity HDR DNVS problem.

## 3 METHOD

**Problem** In HDR DNVS, we aim to learn a HDR 3D model $\mathcal{F}_h$ for a target dynamic scene $G$, $\mathcal{F}_h : (t', V') \rightarrow \mathbf{I}^h_{t',V'}$, that would render an HDR image $\mathbf{I}^h_{t',V'}$ for any timestamp $t'$ and viewpoint $V'$. To that end, we capture a set of multi-exposure LDR training images $\mathbf{I}^l = \{\mathbf{I}^l_1, \cdots, \mathbf{I}^l_t, \cdots, \mathbf{I}^l_T\}$, each $\mathbf{I}^l_t$ associated with the exposure time $e_t$ sampled from the choices $E = \{e_1, \cdots, e_P\}$, and the camera viewpoint $V_t$ selected from $Q$ distinct viewpoints $\mathbf{V} = \{V_1, \cdots, V_Q\}$ where $T$ denotes the total timesteps and $P$ the number of exposure time choices. We may have *optional* access to coupled HDR training data $\mathbf{I}^h = \{\mathbf{I}^h_1, \cdots, \mathbf{I}^h_T\}$.

### 3.1 HDR-4DGS OVERVIEW

The HDR DNVS problem comes with additional complexity of time-evolving structures and illumination as compared to HDR NVS. To overcome that, we introduce HDR-4DGS, a Gaussian Splatting-based framework that reconstructs 4D spatiotemporal HDR radiance fields. HDR-4DGS is composed of a generic dynamic scene representation model and a novel dynamic tone-mapping mechanism. An overview is depicted in Fig. 1.

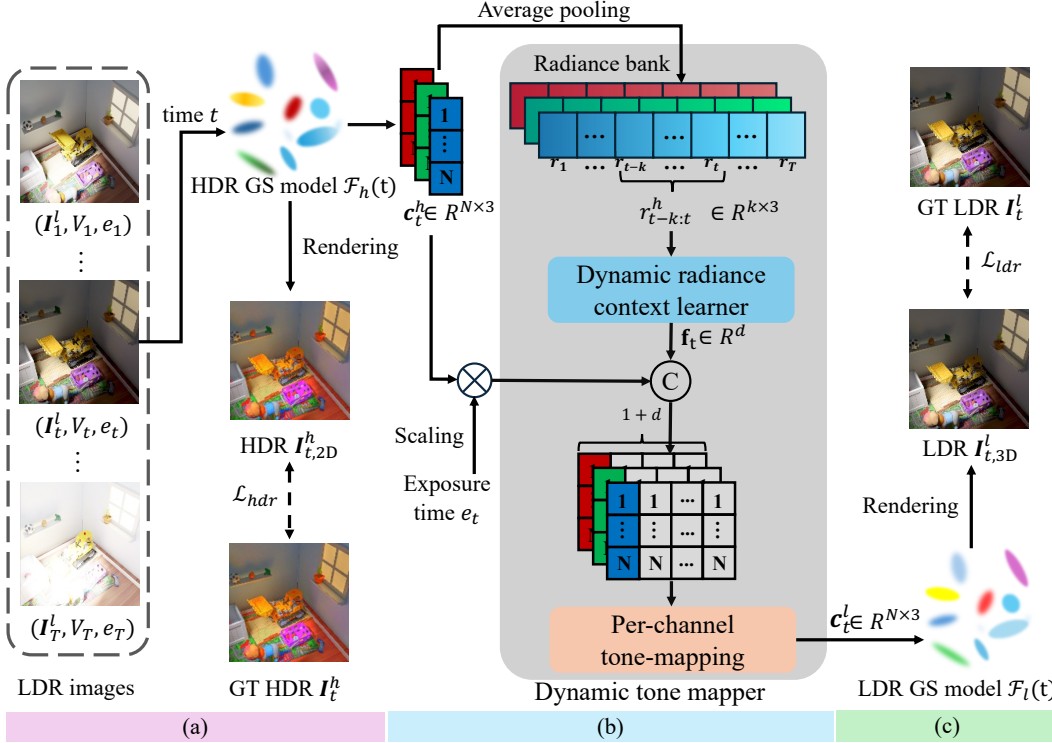

Figure 1: **Overview of HDR-4DGS**. (a) Input data and scene representation; (b) Our proposed Dynamic Tone Mapper (DTM) for temporally adaptive HDR–LDR translation; (c) Loss formulation for joint optimization of geometry, radiance, and tone mapping. $\otimes$ : Dot product. ©: Concatenation.

### 3.2 DYNAMIC SCENE REPRESENTATION

HDR-4DGS can integrate with existing dynamic scene representation to better capture spatiotemporal variations. Specifically, we adopt the 4D Gaussian Splatting (4DGS) framework (Yang et al., 2024) due to its conceptual elegance and coherent formulation of dynamic scenes.

4DGS extends the formulation of 3DGS (Kerbl et al., 2023) by introducing a temporal dimension, allowing pixel observations $\mathbf{I}$ to depend not only on spatial coordinates $(u, v)$ in the image plane but also on an explicit timestamp $t$. This reformulates the original 3DGS framework as:

$$\mathbf{I}(u,v,t) = \sum_{i=1}^{N} p_i(t)p_i(u,v|t)\alpha_i c_i \prod_{j=1}^{i-1}(1 - p_j(t)p_j(u,v|t)\alpha_j), \tag{1}$$

where $p_i(t)$ is the marginal probability over time $t$, $p_i(u,v|t)$ is the conditional spatial probability given $t$, such that $p_i(u,v,t) = p_i(u,v|t) \cdot p_i(t)$, and $N$ denotes the number of Gaussian points. Time and space are treated equally to form a unified 4D Gaussian model, where each Gaussian's mean is denoted by $\mu = (\mu_x, \mu_y, \mu_z, \mu_t)$, and its covariance matrix is defined by $\Sigma = RSS^{\top}R^{\top}$ with appropriately extended rotation matrix $R$ and scaling matrix $S$. The marginal $p(t)$ follows a one-dimensional Gaussian distribution: $p(t) = \mathcal{N}(t; \mu_t, \Sigma_t)$, where $\mathcal{N}(\cdot)$ denotes a normal distribution.

Additionally, 4DGS incorporates a 4D extension of spherical harmonics (SH) to represent the temporal evolution of appearance. The view-dependent color $\mathbf{c}_i$ is modeled using a combination of 4D spherical harmonics, constructed by integrating traditional SH with Fourier series. This design enables dynamic modeling of radiance variations across time, supporting the construction of a radiance bank that facilitates HDR-LDR translation.

In our context, we extend the original color representation space of 4DGS from LDR colors to HDR colors, enabling the accurate synthesis of high-fidelity radiance fields that capture a broader range of luminance variations inherent in real-world dynamic scenes.

### 3.3 DYNAMIC TONE MAPPER

To address the challenge of maintaining temporal radiance consistency in dynamic scenes, we propose a novel dynamic tone mapper (DTM), as shown in Fig. 1(b). DTM explicitly connects the HDR and LDR domains by dynamically adapting per-channel tone-mapping functions in response to evolving temporal radiance patterns. Given a timestamp $t$, exposure time $e_t$, and current HDR color attributes $\mathbf{c}_t^h \in \mathbb{R}^{N \times 3}$, DTM translates HDR colors into their LDR counterparts:

$$\mathbf{c}_t^l = \text{DTM}(\mathbf{c}_t^h, e_t, t) \tag{2}$$

where $\mathbf{c}_t^l \in \mathbb{R}^{N \times 3}$ denotes the resulting tone-mapped LDR colors.

Leveraging 4DGS's explicit radiance modeling through 4DSH, DTM first constructs a radiance bank by storing per-timestamp mean HDR color statistics. The radiance signature $\mathbf{r}_t^h$ for each timestamp is calculated as the average over all $N$ Gaussian points:

$$\mathbf{r}_t^h = \frac{1}{N} \sum_{i=1}^{N} \mathbf{c}_{i,t}^h \in \mathbb{R}^3. \tag{3}$$

A sliding window of $k$ previous frames collects the sequence $\{\mathbf{r}_{t-k:t}^h\}$, which is processed by a *Dynamic Radiance Context Learner* (DRCL) to generate a radiance context embedding:

$$\mathbf{f}_t = \text{DRCL}(\mathbf{r}_{t-k:t}^h) \in \mathbb{R}^d, \tag{4}$$

where $d$ denotes the dimension of this context embedding. DRCL can be generally realized by any existing sequence model such as RNN (Salehinejad et al., 2017), LSTM (Staudemeyer & Morris, 2019), GRU (Cho et al., 2014) or Transformer (Vaswani et al., 2017).

To perform adaptive tone mapping, we concatenate the scaled HDR colors (converted to the logarithmic domain) with the exposure time and radiance context embedding:

$$\mathbf{c}_t^l = g_\theta([\log\ \mathbf{c}_t^h + \log\ e_t, \mathbf{f}_t]), \tag{5}$$

where $g_\theta$ is the per-channel tone-mapping function and $[\cdot]$ denotes concatenation. Scaling HDR colors by exposure time aligns with the principles of the CRF, which models the mapping between scene radiance and observed intensity as exposure-dependent. This normalization ensures consistent appearance modeling across varying exposure settings and facilitates stable learning. By incorporating $\mathbf{f}_t$, DTM enables radiance context-aware HDR-to-LDR translation, improving radiance consistency across time in dynamic scenes.

### 3.4 MODEL OPTIMIZATION

The overall objective function used to optimize HDR-4DGS is defined as:

$$\mathcal{L}_{\text{total}} = \mathcal{L}_{\text{ldr}} + \alpha \mathcal{L}_{\text{hdr}}, \tag{6}$$

where $\mathcal{L}_{\text{ldr}}$ denotes the loss computed in the LDR domain, and $\mathcal{L}_{\text{hdr}}$ refers to the HDR reconstruction loss. The weighting factor $\alpha$ is set to zero if HDR ground truth is unavailable during training.

To mitigate overfitting associated with directly applying 3D tone mapping on HDR Gaussian fields (Liu et al., 2025a), we adopt an extra pixel-level supervision in addition to existing ray-level supervision over the dynamic tone mapper, which introduces additional constraints to facilitate robust CRF learning: $\mathbf{I}_{t,2}^l = g_\theta([\log\ \mathbf{I}_{t,2D}^h + \log\ e_t, \mathbf{f}_t])$, where $\mathbf{I}_{t,2D}^h$ is the HDR image rasterized by the HDR Gaussian model, and $\mathbf{I}_{t,2D}^l$ is the tone-mapped LDR image at time $t$. During training, we supervise both the tone-mapped LDR image $\mathbf{I}_{t,2D}^l$ and the LDR image $\mathbf{I}_{t,3D}^l$ rendered directly from the LDR model. This dual supervision improves the learned tone mapper's generalization capability, as validated in our ablation (Tab. 5).

We define the image reconstruction loss as a weighted combination of $L_1$ and D-SSIM loss (Huang et al., 2022; Nilsson & Akenine-Möller, 2020):

$$\mathcal{L}(\mathbf{I}_1, \mathbf{I}_2) = (1 - \lambda)\mathcal{L}_1(\mathbf{I}_1, \mathbf{I}_2) + \lambda \mathcal{L}_{\text{D-SSIM}}(\mathbf{I}_1, \mathbf{I}_2), \tag{7}$$

where $\mathbf{I}_1/\mathbf{I}_2$ are paired and $\lambda$ balances their contributions. Accordingly, the losses are defined as:

$$\mathcal{L}_{\text{ldr}} = \mathcal{L}(\mathbf{I}_{t,2D}^l, \mathbf{I}_t^l) + \mathcal{L}(\mathbf{I}_{t,3D}^l, \mathbf{I}_t^l), \quad \mathcal{L}_{\text{hdr}} = \mathcal{L}(\hat{\mathbf{I}}_{t,2D}^h, \hat{\mathbf{I}}_t^h), \tag{8}$$

Table 1: Results on HDR-4D-Syn. *: HDR only supervision; †: LDR+HDR supervision.

| Row | Method | Supervision | HDR | | | LDR | | | Training time (min) | Inference speed (fps) |
|-----|--------|-------------|------|------|--------|------|------|--------|------|------|
| | | | PSNR↑ | SSIM↑ | LPIPS↓ | PSNR↑ | SSIM↑ | LPIPS↓ | | |
| 1 | HexPlane | LDR | - | - | - | 14.20 | 0.564 | 0.551 | 28.45 | 1.60 |
| 2 | HexPlane* | HDR | 24.89 | 0.771 | 0.377 | - | - | - | 59.64 | 1.94 |
| 3 | 4DGS | LDR | - | - | - | 13.92 | 0.549 | 0.281 | 116.25 | 75.82 |
| 4 | 4DGS* | HDR | 22.40 | 0.650 | 0.345 | - | - | - | 44.58 | 172.61 |
| 5 | HDR-NeRF | LDR | 8.54 | 0.062 | 0.552 | 21.66 | 0.664 | 0.553 | 212.83 | 0.061 |
| 6 | HDR-GS | LDR | 4.64 | 0.158 | 0.645 | 6.45 | 0.272 | 0.423 | **13.88** | 380.38 |
| 7 | HDR-GS† | LDR+HDR | 14.33 | 0.360 | 0.482 | 10.84 | 0.372 | 0.378 | 22.50 | 255.21 |
| 8 | HDR-HexPlane | LDR | 14.70 | 0.649 | 0.287 | 32.59 | 0.912 | 0.145 | 37.83 | 1.61 |
| 9 | HDR-HexPlane† | LDR+HDR | 29.30 | 0.844 | 0.223 | 31.09 | 0.896 | 0.185 | 54.31 | 1.33 |
| 10 | **HDR-4DGS (Ours)** | LDR | 25.88 | 0.865 | **0.076** | **33.16** | **0.949** | **0.055** | 69.38 | 40.80 |
| 11 | **HDR-4DGS† (Ours)** | LDR+HDR | **30.40** | **0.914** | 0.097 | 30.69 | 0.927 | 0.097 | 76.86 | 48.63 |

Figure 2: Visual comparison of HDR DNVS on HDR-4D-Syn.

where $\mathbf{I}_t^l$ is the ground-truth LDR image. The tone-mapped HDR images $\hat{\mathbf{I}}_{t,2D}^h$ and $\hat{\mathbf{I}}_t^h$ are derived from rendered HDR image $\mathbf{I}_{t,2D}^h$ and ground-truth HDR image $\mathbf{I}_t^h$, respectively, using $\mu$-law compression: $\hat{\mathbf{I}}^h = \frac{\log(1+\mu \cdot \text{norm}(\mathbf{I}^h))}{\log(1+\mu)}$ where $\mu$ is a compression factor and $\text{norm}(\cdot)$ is min-max normalization. This transformation aligns HDR and LDR domains for consistent comparison. Notably, we choose the LDR images $\mathbf{I}_{t,3D}^l$ rasterized by Gaussian Splatting as our final LDR rendering results.

## 4 EXPERIMENTS

**Datasets.** Due to no benchmarks for HDR DNVS, we introduce two complementary datasets: *HDR-4D-Syn* and *HDR-4D-Real*. HDR-4D-Syn consists of 8 synthetic dynamic scenes adapted from the dataset by Wu et al. (2024a). It features multi-exposure video sequences captured under varying exposure settings, accompanied by synchronized multi-view LDR video streams. Corresponding HDR ground truth frames are re-synthesized to ensure high-fidelity supervision and evaluation. The real-world dataset, HDR-4D-Real, captures 4 dynamic indoor scenes in real-world settings. Videos are recorded under three distinct exposure times using six synchronized iPhone 14 Pro devices. Ground truth HDR images are generated UltraFusion (Chen et al., 2025), ensuring realistic and high-quality HDR reconstructions. Further details are provided in Appendix A.1.

**Implementation details.** HDR-4DGS is trained with the Adam optimizer using the same parameters as 4DGS (Yang et al., 2024). For our dynamic tone mapper, the learning rate is set to $5 \times 10^{-4}$, and the dimension of temporal radiance context features is set to 2. We adopt the same structure of per-channel tone-mapping functions in our dynamic tone mapper as HDR-GS (Cai et al., 2024) and GRU (Cho et al., 2014) is adopted to implement the dynamic radiance context learner by default. For equation 7, $\lambda$ is set to 0.2, and $\alpha$ is set to 0.6 if HDR ground truth is available.

**Evaluation metrics.** We adopt the PSNR and SSIM as quantitative metrics, LPIPS as an additional perceptual metric. Following prior works (Cai et al., 2024; Huang et al., 2022; Liu et al., 2025a), we apply Photomatix Pro (HDRsoft Team, 2025) to convert HDR images into displayable LDR images for qualitative visualization and fair comparison. Futhermore, training time and inference speed (fps) are reported. *Results are averaged over all scenes.*

### 4.1 QUANTITATIVE EVALUATION

**Competitors.** We compare HDR-4DGS with latest alternatives: HexPlane (Cao & Johnson, 2023), 4DGS (Yang et al., 2024), HDR-HexPlane (Wu et al., 2024a), HDR-NeRF (Huang et al., 2022)

Table 2: Results on HDR-4D-Real. *: HDR only supervision; †: LDR+HDR supervision.

| Row | Method | Supervision | HDR | | | LDR | | | Training time (min) | Inference speed (fps) |
|---|---|---|---|---|---|---|---|---|---|---|
| | | | PSNR↑ | SSIM↑ | LPIPS↓ | PSNR↑ | SSIM↑ | LPIPS↓ | | |
| 1 | HexPlane | LDR | - | - | - | 13.82 | 0.551 | 0.576 | **25.50** | 0.44 |
| 2 | HexPlane* | HDR | 32.76 | 0.893 | 0.242 | - | - | - | 28.24 | 0.49 |
| 3 | 4DGS | LDR | - | - | - | 7.99 | 0.072 | 0.620 | 42.50 | **290.57** |
| 4 | 4DGS* | HDR | 7.85 | 0.220 | 0.534 | - | - | - | 40.75 | 307.85 |
| 5 | HDR-NeRF | LDR | 14.60 | 0.711 | 0.411 | 8.326 | 0.029 | 0.943 | 212.25 | 0.17 |
| 6 | HDR-GS | LDR | 13.27 | 0.783 | 0.261 | 20.52 | 0.840 | 0.148 | 38.25 | 73.30 |
| 7 | HDR-GS† | LDR+HDR | 29.40 | **0.936** | **0.097** | 20.85 | 0.834 | 0.182 | 56.50 | 64.88 |
| 8 | HDR-HexPlane | LDR | 9.306 | 0.672 | 0.353 | 27.44 | 0.748 | 0.353 | 36.98 | 0.35 |
| 9 | HDR-HexPlane† | LDR+HDR | **33.03** | 0.904 | 0.192 | 28.12 | 0.767 | 0.307 | 44.43 | 0.24 |
| 10 | **HDR-4DGS (Ours)** | LDR | 14.50 | 0.884 | 0.200 | 26.88 | 0.825 | 0.221 | 98.75 | 35.27 |
| 11 | **HDR-4DGS† (Ours)** | LDR+HDR | 25.13 | 0.909 | 0.162 | **30.69** | **0.927** | **0.097** | 76.86 | 48.63 |

Figure 3: Visual comparison of HDR DNVS on HDR-4D-Real.

and HDR-GS (Cai et al., 2024). HDR-HexPlane preliminarily explores dynamic HDR synthesis by extending HexPlane with exposure conditioning and a static Sigmoid tone mapper. However, it lacks HDR supervision and omits explicit evaluation of HDR outputs, all of which have been addressed here for both fair comparison and completeness.

Although HDR-GS and HDR-NeRF were originally designed for static scenes, we adapt them on dynamic scenes to underscore the importance of explicit spatial modeling in dynamic HDR reconstruction. All models are tested with their official implementations to ensure optimal performance.

**Results.** Tab. 1 and Tab. 2 present the results for both LDR and HDR DNVS on HDR-4D-Syn and HDR-4D-Real, respectively. HDR-NeRF encounters numerical instability during training when both LDR and HDR supervision are applied simultaneously, frequently leading failed optimization. Meanwhile, 4DGS struggles to reconstruct HDR scenes on HDR-4D-Real under HDR-only supervision, as ground-truth HDR images generated via 2D methods lack multi-view consistency. We highlight the following key findings, focusing on HDR DNVS performance:

**(I) Superior HDR DNVS quality.** HDR-4DGS consistently outperforms all competing methods, including HDR-NeRF, HDR-GS, and HDR-HexPlane for HDR DNVS on HDR-4D-Syn, owing primarily to our explicit spatiotemporal HDR radiance modeling and the integration of a dynamic tone mapping mechanism that preserves temporal radiance coherence across views and exposures. In contrast, HDR-GS and HDR-NeRF rely on static tone mappers, failing to accurately bridge the HDR and LDR domains; while HDR-HexPlane achieves higher PSNR on HDR-4D-Real, we attribute this largely to the noise in HDR ground truth and the known limitations of PSNR as a metric, particularly its tendency to favor overly smooth or blurred reconstructions (Li et al., 2020), as visually demonstrated in Fig. 9. As shown in Fig. 3 and videos in the supplementary material, HDR-4DGS preserves significantly sharper spatial structure and more accurate color details than all competitors. Although HDR-4DGS exhibits structural degradation in movable regions, we attribute this limitation to the inherent expressive capacity of our base representation model, 4DGS, rather than instability in HDR-4DGS's handling of dynamic ranges. For further visual evidence, please refer to Fig. 10 and Fig. 11 in the Appendix where the vanilla 4DGS (Yang et al., 2024) suffers from even more severe structural distortion and geometry loss. In contrast, our HDR-4DGS alleviates these degradations to a noticeable extent, supporting the effectiveness of the proposed HDR modeling strategy.

**(II) Effectiveness of dynamic tone mapping.** Our dynamic tone mapper plays a central role in enhancing HDR synthesis. By modeling exposure-dependent radiance dynamics with temporal con-

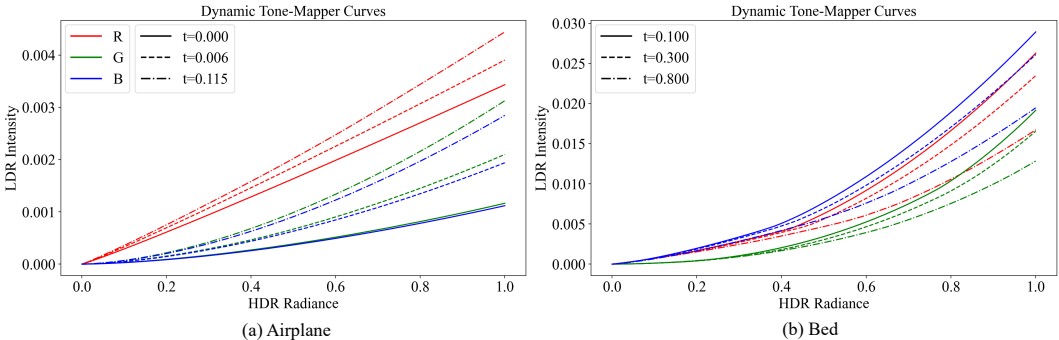

(a) Airplane                    (b) Bed

Figure 4: Temporal variation with learned tone mapping patterns by DTM in two scenes.

text, it enables precise HDR-LDR translation and promotes stability during training and inference. This proves essential not only for high-fidelity HDR rendering but also for generalizing under varying lighting conditions.

**(III) Training flexibility and robustness.** HDR-4DGS exhibits strong resilience across diverse supervision settings. While both HDR-4DGS and HDR-HexPlane benefit from joint LDR-HDR supervision, HDR-4DGS maintains competitive performance even with LDR-only supervision. This highlights the ability of our dynamic tone mapper and radiance context learner to extract and exploit temporal radiance correlations, enabling accurate HDR reconstruction even without HDR labels.

**(IV) Practical efficiency.** In addition to rendering quality, HDR-4DGS delivers substantial efficiency improvements. It achieves up to $36\times$ and $200\times$ faster inference compared to HDR-HexPlane while preserving competitive training times on HDR-4D-Syn and HDR-4D-Real, respectively. This efficiency gain is critical for scaling HDR synthesis to real-world, dynamic scenarios where runtime performance is a key constraint. Note that adding HDR supervision leads to faster inference for HDR-4DGS (Row 10 v.s. 11 in Tab. 1 and Tab. 2) since using HDR signal would effectively compress the model, as there is no need for the model to generate redundant Guassian points to approximate the transformation from LDR to HDR.

Overall, the results confirm that HDR-4DGS sets a new state of the art in HDR DNVS by jointly optimizing HDR fidelity, dynamic tone adaptation, and inference efficiency both on synthetic and real datasets.

## 4.2 DYNAMICS IN TONE MAPPING CAPTURED

To verify that our DTM learns *dynamic* rather than static tone mapping, we analyze scenes with pronounced motion-induced brightness changes (*e.g.*, Airplane and Bed). After training HDR-4DGS, we extract the learned DTM module along with the radiance bank, both jointly define a dynamic mapping from HDR radiance to LDR counterpart, with condition on the time. To assess the dynamics of this learned mapping, we then sample different timestamps and visualize the corresponding tone-mapping curves over the HDR intensity, ass illustrated in Fig. 4.

We observe that **(I)** all curves are monotonically increasing (consistent with CRF definition (Huang et al., 2022)) yet exhibit scene-specific patterns, indicating adaptive tone reproduction; **(II)** In the Airplane scene, the red channel curve consistently lies above green and blue, reflecting the scene's dominant reddish tone. As the airplane moves from shadows to brighter regions, curves shift upward over time, tracking increasing luminance; **(III)** In the Bed scene, curves gradually descend as the bed unfolds and ambient lighting dims. These temporal dynamics confirm DTM's ability to adapt tone mapping to evolving lighting conditions and scene content.

## 4.3 QUALITATIVE RESULTS

While PSNR, SSIM, and LPIPS are commonly used metrics, they may not fully capture perceptual quality, necessitating qualitative visual evaluation. As shown in Fig. 2, HDR-HexPlane fails to reconstruct extreme radiance details, while HDR-4DGS successfully captures more intricate structures. From Fig. 3, it can be observed that HDR-GS struggles to reconstruct dynamic objects, and

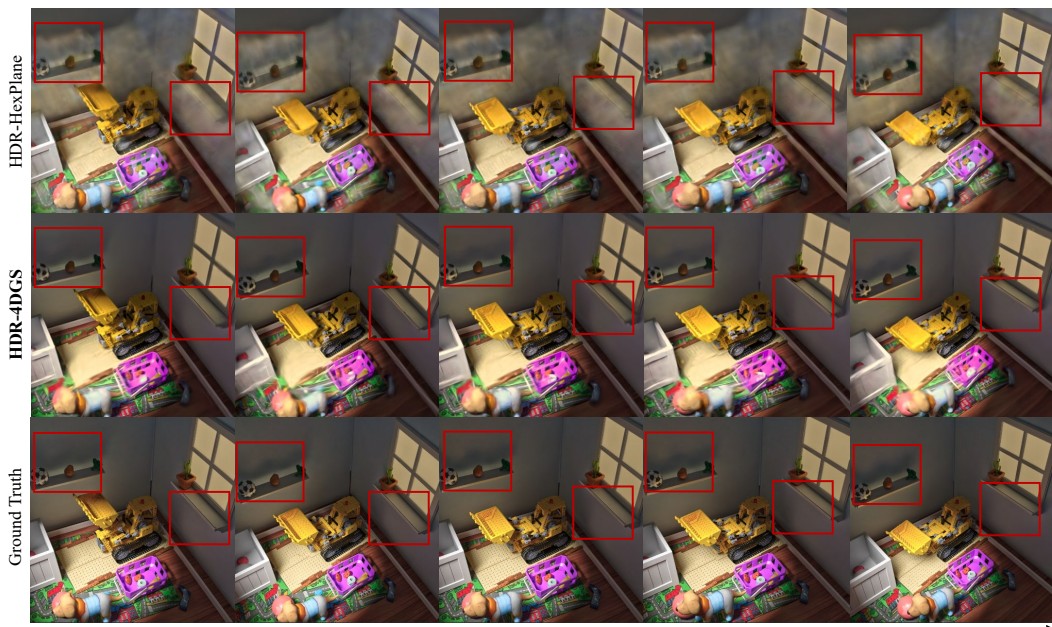

Figure 5: Comparison of HDR renderings' temporal radiance variations.

HDR-HexPlane tends to produce over-smoothed results. In contrast, HDR-4DGS effectively represents the dynamic scene while preserving finer details. Meanwhile, temporal radiance consistency is also challenged by HDR-HexPlane, while HDR-4DGS synthesizes spatiotemporally coherent HDR results, as illustrated in Fig. 5. Please refer to Appendix A.2 for more visualization comparisons.

## 4.4 ABLATION STUDY

We conduct ablation studies on the HDR-4D-Syn dataset (see Appendix A.2 for more experiments).

**Effect of joint HDR reconstruction.** To validate the effect of our DTM, we train 4DGS on single-exposure LDR images (exposure time 2.0s) and convert the synthesized LDR images of novel views to HDR images with a variety of existing Single HDR Image Reconstruction models: EIN (Liu et al., 2025b), IntrinsicHDR (Dille et al., 2024), KPNet (Wang et al., 2024a), KUNet (Wang et al., 2022).

Tab. 3 shows that the two-stage pipeline significantly underperforms our HDR-4DGS since converting single-exposure LDR inputs to HDR is inherently ill-posed where the missing radiance information cannot be reliably recovered in isolation from the scene reconstruction process. Our integrated approach instead jointly optimizes radiance representation and novel-view synthesis, establishing its necessity and irreplaceability for high-fidelity HDR scene reconstruction.

Table 3: Results of 4DGS with independent HDR.

| Method | HDR | | |
| --- | --- | --- | --- |
| | PSNR ↑ | SSIM ↑ | LPIPS ↓ |
| 4DGS + EIN (Liu et al., 2025b) | 17.75 | 0.770 | 0.185 |
| 4DGS + IntrinsicHDR (Dille et al., 2024) | 11.63 | 0.673 | 0.251 |
| 4DGS + KPNet (Wang et al., 2024a) | 20.92 | 0.813 | 0.197 |
| 4DGS + KUNet (Wang et al., 2022) | 19.00 | 0.715 | 0.172 |
| HDR-4DGS (Ours) | **25.88** | **0.865** | **0.076** |

**Effect of dynamic tone mapper.** We conduct comparative experiments against an MLP variant (part of HDR-GS (Cai et al., 2024) and HDR-NeRF (Huang et al., 2022), used for static tone-mapping), two classical tone mappers Durand (Durand & Dorsey, 2000) and Reinhard (Reinhard et al., 2023) and the static tone mapper adopted by HDR-HexPlane. Additionally, ablation studies are performed to assess the contribution of extra pixel-level supervision (see Sec. 3.4). Key findings are: **(I)** Tab. 4 reveals that when replacing our dynamic tone mapper with other existing counterparts, we observe significant performance degradation. This confirms that explicit modeling of temporal radiance variations through our dynamic tone mapper is critical for maintaining HDR fidelity while existing mappers are inferior in doing that. Further, complementary ablation studies with respect to the scene representation verify that the observed performance gains are primarily attributable to our dynamic tone mapper, rather than 4DGS alone. **(II)** Disabling the pixel-level supervision leads to performance reduction (PSNR decreases by 1.03 dB), as shown in Tab. 5. This demonstrates

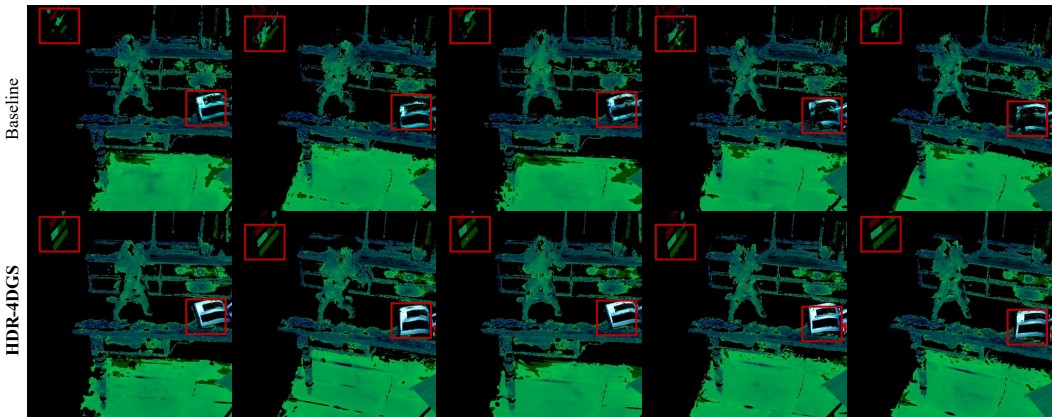

Figure 6: Continuous radiance variations comparison of HDR DNVS. Photomatix Pro (HDRsoft Team, 2025) is used to facilitate the comparison of radiance transition.

that joint optimization with both ray-level and pixel-level constraints provides essential supervisory signals for learning physically plausible tone mapping operators.

In addition, to rigorously evaluate the efficiency of our proposed DTM in preserving continuous radiance variations, we conduct ablation studies by substituting the DTM with an MLP baseline. Specifically, we synthesize temporally coherent HDR sequences for a dynamic scene using both architectures, as shown in Fig. 6 (also see Fig. 17) where our HDR-4DGS with DTM achieves superior temporal coherence in radiance transitions compared to baseline approach, validating the effectiveness of our DTM in preserving HDR coherence in dynamic domain.

Table 4: Ablation on dynamic tone mapping.

| Method | PSNR↑ | SSIM↑ | LPIPS↓ |
|---|---|---|---|
| Reinhard (Reinhard et al., 2023) | 22.10 | 0.812 | 0.210 |
| Durand (Durand & Dorsey, 2000) | 22.85 | 0.825 | 0.195 |
| MLP (HDR-GS) | 23.92 | 0.841 | 0.142 |
| **DTM (Ours)** | **25.88** | **0.865** | **0.076** |

Table 5: Analysis of pixel-level supervision.

| Pixel-level Supervision | HDR PSNR↑ | HDR SSIM↑ | HDR LPIPS↓ |
|---|---|---|---|
| No | 24.85 | 0.853 | 0.169 |
| Yes | **25.88** | **0.865** | **0.076** |

**Temporal length.** As described in Sec. 3.3, our proposed dynamic tone mapper (DTM) extracts radiance cues from the past $k$ timestamps in the radiance bank using a sliding window and a dynamic radiance context learner. The hyper-parameter $k$ governs the trade-off between temporal context coverage and computational efficiency: larger $k$ captures extended radiance dynamics but risk redundancy and noise sensitivity (*e.g.*, scenes with large, rapid motions or complex illumination dynamics), while smaller $k$ prioritizes immediate temporal cues at the cost of modeling complex patterns (*e.g.*, scenes characterized by minor motion or slow illumination variation). Through ablation experiments with $k$, as shown in Tab. 6, we find that $k = 20$ achieves optimal performance, balancing responsiveness to dynamic scenes with practical efficiency constraints.

Table 6: Analysis of the temporal context length.

| Row | $k$ | HDR PSNR↑ | HDR SSIM↑ | HDR LPIPS↓ |
|---|---|---|---|---|
| 1 | 5 | 24.74 | 0.852 | 0.092 |
| 2 | 10 | 24.76 | 0.851 | 0.098 |
| 3 | 20 | **25.88** | **0.856** | **0.076** |
| 4 | 30 | 24.29 | 0.825 | 0.094 |

## 5 CONCLUSION

This work introduces *High Dynamic Range Dynamic Novel View Synthesis* (HDR DNVS), addressing a critical limitation in prior HDR synthesis restricted to static scenes. We present HDR-4DGS, a novel framework designed to reconstruct temporally coherent HDR radiance fields and dynamic geometry from sparse, time-varying LDR observations. The key of HDR-4DGS lies in its dynamic tone mapping module, which leverages a radiance bank and a dynamic radiance context learner to drive per-channel tone-mapping functions that adaptively bridge HDR and LDR domains across time. Extensive experiments demonstrate that HDR-4DGS significantly outperforms prior methods in HDR rendering fidelity, temporal radiance consistency, and computational efficiency.

## 6 Ethics Statement

This work is focused on advancing the technical capabilities of dynamic scene reconstruction and HDR rendering from standard LDR video inputs. The proposed method, HDR-4DGS, is intended for research and non-malicious applications such as virtual reality, cinematic content creation, and immersive telepresence.

The synthetic data used in our experiments are generated in controlled simulation environments, and the real-world scenes in our dataset were captured with the informed consent of all participants and property owners. No personally identifiable information is included in the released data. We do not anticipate direct negative societal impacts from this research; however, as with any novel view synthesis technology, potential misuse (*e.g.*, generating misleading visual content) could arise if deployed without appropriate safeguards. We encourage responsible use and advocate for transparency in synthetic media generation.

## 7 Reproducibility Statement

We are committed to the reproducibility of HDR-4DGS. The complete code will be publicly released upon final acceptance of this paper. To facilitate verification prior to code release, we provide a thorough description of our method in Sec. 3 and comprehensive implementation details in Sec. 4 and Appendix A.2. Together, these sections cover all essential components of HDR-4DGS, enabling independent replication of our results.

## 8 Acknowledgments

This work was supported by the Key Research and Development Program of Jiangsu Province (Industry Foresight and Key Core Technologies) under Grant No. BE2023008-2, and by the UKRI-AHRC CoSTAR National Lab for Creative Industries Research and Development (AH/Y001060/1).

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

# A APPENDIX

## A.1 DATASETS

This section elaborates on the HDR-4D-Syn and HDR-4D-Real datasets introduced in this work. As summarized in Tab. 7, the HDR-4D-Syn dataset is synthesized with Blender which comprises four monocular acquisition configurations (Airplane, Deer, Lego and Tank) where distinct exposure times are applied across viewpoints, and four multi-view stereo configurations (Hook, Jump, Mutant and Standup) characterized by multi-exposure LDR images capture per viewpoint. This design enables comprehensive evaluation of dynamic scene reconstruction under varying lighting conditions and camera setups. Notably, we explicitly use different exposure times rather than exposure value to render multi-exposure images, which enables precise control over radiance sampling intervals, aligning with recent paper conventions. Fig. 7 shows some example HDR images and LDR images with different exposure times of different scenes in HDR-4D-Syn.

Table 7: Statistics of HDR-4D-Syn, HDR / LDR means the number of HDR / LDR images.

| Scenes | Training | | | Testing | | | Cameras | Resolution | Exposure Time (s) |
|--------|--------|-----|-----|--------|-----|-----|---------|------------|----------|
|        | Frames | HDR | LDR | Frames | HDR | LDR |         |            |          |
| Airplane | 280 | 280 | 280 | 70 | 70 | 70 | 1 | 800×800 | 0.125/2/32 |
| Deer | 80 | 80 | 80 | 20 | 20 | 20 | 1 | 800×800 | 0.125/2/32 |
| Hook | 28 | 280 | 840 | 4 | 40 | 120 | 10 | 800×800 | 0.125/2/32 |
| Jump | 21 | 210 | 630 | 3 | 30 | 90 | 10 | 800×800 | 0.125/2/32 |
| Lego | 240 | 240 | 240 | 60 | 60 | 60 | 1 | 800×800 | 0.125/2/32 |
| Mutant | 135 | 405 | 405 | 14 | 42 | 42 | 3 | 800×800 | 0.125/2/32 |
| Standup | 51 | 255 | 765 | 8 | 40 | 120 | 5 | 800×800 | 0.125/2/32 |
| Tank | 136 | 136 | 136 | 34 | 34 | 34 | 1 | 800×800 | 0.125/2/32 |

As for the HDR-4D-Real dataset, to ensure capture accuracy and stability, we employed a fixed multi-camera setup with six cameras securely mounted on tripods. Prior to introducing any dynamic elements, we captured multi-view static images of the empty scene. These static images were processed using COLMAP (Schonberger & Frahm, 2016) to compute precise intrinsic and extrinsic camera parameters, establishing a stable and accurate spatial reference frame. Crucially, all camera positions and optical parameters were rigorously maintained throughout the entire dynamic capture sequence, eliminating the need for recalibration and guaranteeing consistent geometric alignment across all timestamps. For each camera and each timestamp within the dynamic sequences, we captured three images with different exposure times (*e.g.*, 1/120s, 1/50s, and 1/20s), enabling the subsequent synthesis of HDR ground truth. Finally, we adopt UltraFusion (Chen et al., 2025) to synthesize the corresponding HDR ground truth from multi-exposure LDR images. Tab. 8 summarizes the statistics of HDR-4D-Real, and Fig. 8 shows some example HDR images and LDR images with different exposure times of different scenes.

**Remark.** It is noted that using identical synchronized cameras is indeed common practice in dynamic scene capture, as evidenced by major datasets like the Neural 3D Video Dataset (Li et al., 2022a) (18-21 synchronized cameras), Google Immersive Dataset (Broxton et al., 2020) (46-camera rig), and Technicolor Light Field Dataset (Sabater et al., 2017) (4×4 synchronized array). This approach is both a methodological convention and a practical necessity—synchronization ensures temporal consistency critical for dynamic reconstruction, while identical camera models eliminate radiometric calibration complexities from varying CRFs. Asynchronous capture or mixed camera setups would introduce significant additional challenges, including temporal misalignment artifacts,

Table 8: Statistics of HDR-4D-Real, HDR / LDR means the number of HDR / LDR images.

| Scenes | Training | | | Testing | | | Cameras | Resolution | Exposure Time (s) |
|--------|--------|-----|-----|--------|-----|-----|---------|------------|----------|
|        | Frames | HDR | LDR | Frames | HDR | LDR |         |            |          |
| Bed | 22 | 132 | 317 | 22 | 79 | 79 | 6 | 4032×3024 | 0.004/0.008/0.017 |
| Excavator | 40 | 240 | 576 | 40 | 144 | 144 | 6 | 4032×3024 | 0.002/0.008/0.033 |
| Tank | 20 | 40 | 96 | 20 | 24 | 24 | 2 | 4032×3024 | 0.007/0.02/0.05 |
| Toys | 20 | 40 | 96 | 20 | 24 | 24 | 2 | 4032×3024 | 0.008/0.02/0.1 |

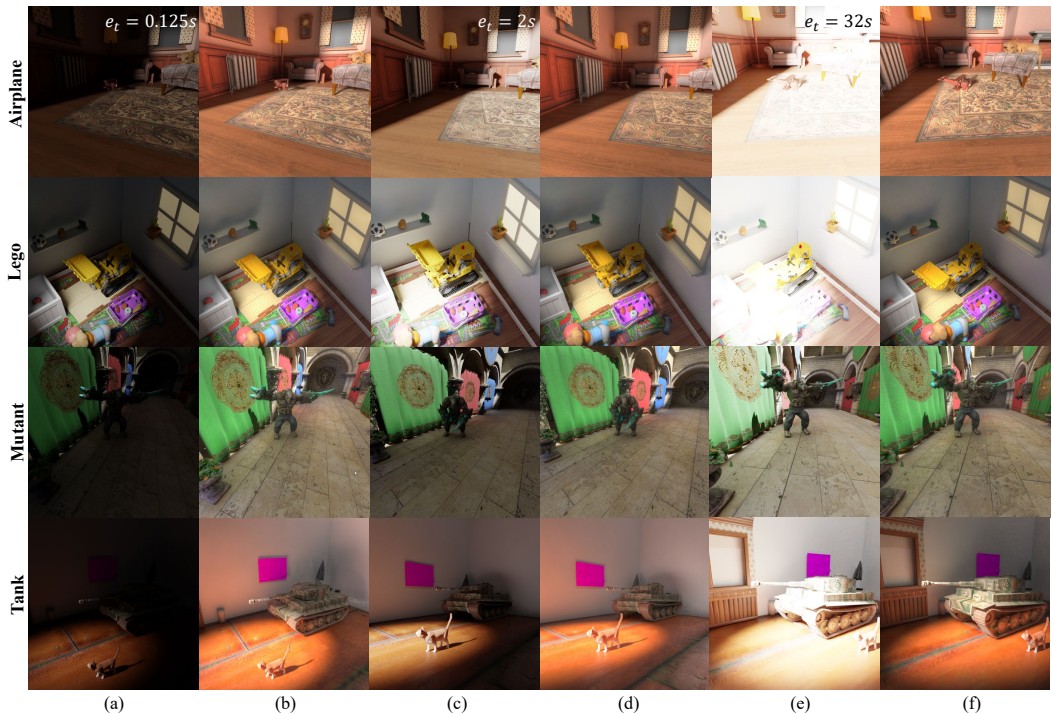

Figure 7: Comparative visualization of LDR and HDR image sequences across dynamic scenes (Airplane, Lego, Mutant, Tank). Each row corresponds to a distinct scene, while columns (a) / (c) / (e) present LDR images captured at varying exposure times and temporal intervals. Columns (b) / (d) / (f) illustrate the HDR counterparts. $e_t$: Exposure time.

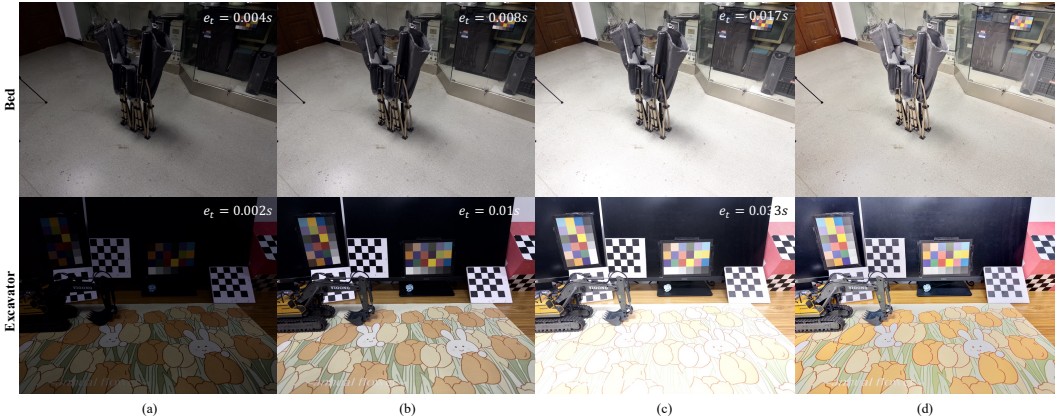

Figure 8: Dataset composition for dynamic scenes (Bed and Excavator) of HDR-4D-Real. Each row corresponds to a distinct scene, with columns (a)–(c) displaying LDR images captured at varying exposure settings but synchronized temporal instances. Columns (d) presents the corresponding HDR images at matching temporal frames. $e_t$: Exposure time.

inconsistent motion blur, and complex cross-camera radiometric calibration that could obscure the core evaluation of HDR reconstruction capabilities. While robustness to such real-world variations represents valuable future work with extra challenges, this work focuses on common practice aligned with established conventions in the field.

## A.2 ADDITIONAL EXPERIMENTS

**Implementation details.** Since Structure-from-Motion (SfM) (Schonberger & Frahm, 2016) struggles to perform reliably on dynamic monocular videos, we adopt different initialization strategies

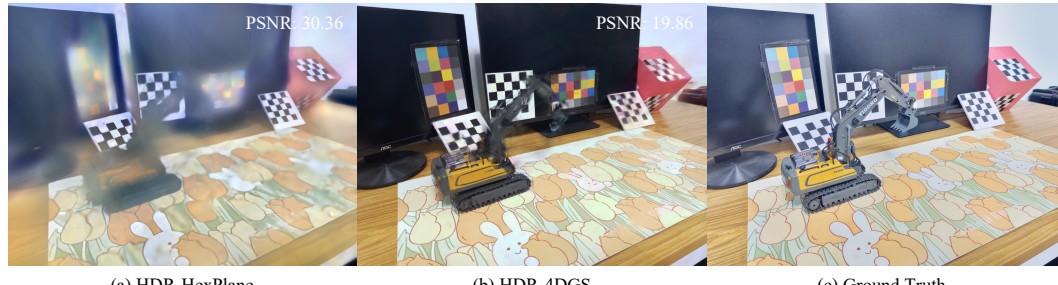

(a) HDR-HexPlane        (b) HDR-4DGS        (c) Ground Truth

Figure 9: PSNR prefers over-smooth or blurry images. HDR images are tone-mapped by Photomatix Pro (HDRsoft Team, 2025).

for synthetic and real-world scenes. Specifically, we randomly initialize $5 \times 10^4$ Gaussians for each synthetic scene, while for real-world scenes, we initialize the Gaussians using a dense reconstructed point cloud and and the input images are downsampled by a factor of 4. In both cases, we maintain the same number of training iterations as 4DGS (Yang et al., 2024). We conducted all the experiments on a single NVIDIA RTX 4090 GPU.

**Ablation study of the design of dynamic radiance context learner.** The temporal radiance variation modeling constitutes a critical component of our dynamic tone mapper, as the temporal coherence of radiance decomposition directly influences HDR rendering performance.

To systematically evaluate architectural suitability, we implement the dynamic radiance context learner with multiple sequence models and quantitatively compare their effectiveness. As shown in Tab. 9, while Transformer-based modules demonstrate theoretical advantages in long-term dependency modeling (Wen et al., 2023), our experiments reveal supe-

Table 9: Analysis of DRCL design.

| Network | HDR | | |
| --- | --- | --- | --- |
| | PSNR ↑ | SSIM ↑ | LPIPS ↓ |
| RNN (Salehinejad et al., 2017) | 25.63 | 0.847 | 0.100 |
| LSTM (Staudemeyer & Morris, 2019) | 25.53 | 0.845 | 0.106 |
| GRU (Cho et al., 2014) | **25.88** | **0.865** | **0.076** |
| Transformer (Vaswani et al., 2017) | 25.06 | 0.817 | 0.101 |

rior performance when employing GRU. This observation aligns with recent findings in sequence feature extraction, where GRU exhibits better efficiency in capturing local temporal patterns (Li et al., 2023).

**Additional visualization results.** Fig. 10 and Fig. 11 provide additional visual comparisons of DNVS performance on the HDR-4D-Syn and HDR-4D-Real datasets. It is observed that 4DGS (Yang et al., 2024) consistently struggles to preserve fine-grained details and frequently exhibits structural degradation in movable regions. In contrast, our HDR-4DGS demonstrably retains superior detail fidelity, even when employing 4DGS as its underlying representation model. This outcome further corroborates the efficacy of our proposed DTM in enhancing detail preservation, as thoroughly discussed in Sec. 3.3.

Fig. 12 and Fig. 13 present additional HDR DNVS rendering results on the HDR-4D-Syn and HDR-4D-Real datasets, respectively. Qualitative comparisons reveal that HDR-4DGS consistently preserves fine structural details and generates visually compelling renderings. In contrast, HDR-HexPlane tends to produce spatially blurred regions, while HDR-GS exhibits significant failure in reconstructing dynamic content, resulting in incomplete renderings.

Similarly, Fig. 14 and Fig. 15 illustrate the LDR DNVS results on the same datasets. HDR-4DGS demonstrates superior color fidelity and detail retention under LDR conditions. Conversely, HDR-HexPlane again suffers from noticeable blurring artifacts, and HDR-GS fails to accurately recover scene chromaticity, further compromising its ability to render dynamic objects with photometric coherence.

Furthermore, accurate rendering of continuous radiance distributions in HDR DNVS is essential for preserving high-fidelity radiance variations across complex lighting conditions, which often occur in the real-world dynamic scenes. As demonstrated in Fig. 16 , our HDR-4DGS with dynamic tone mapper featured by a dynamic radiance context learner achieves superior temporal coherence in radiance transitions compared to baseline approaches, effectively capturing smoother details while maintaining photometric consistency. Meanwhile, Fig. 17 provides additional ablation study re-

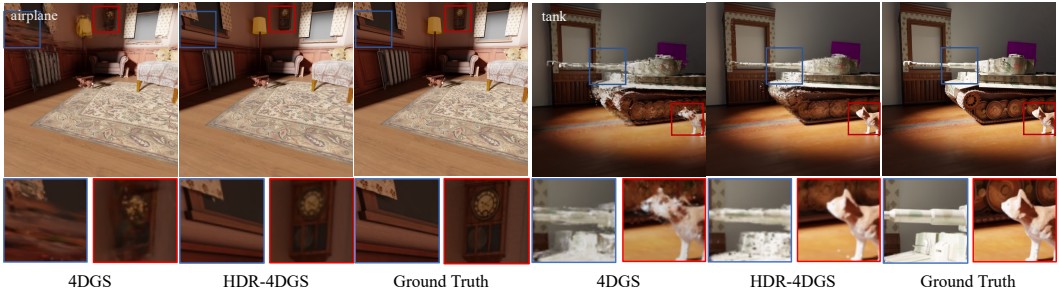

Figure 10: Additional visual comparison of DNVS on HDR-4D-Syn.

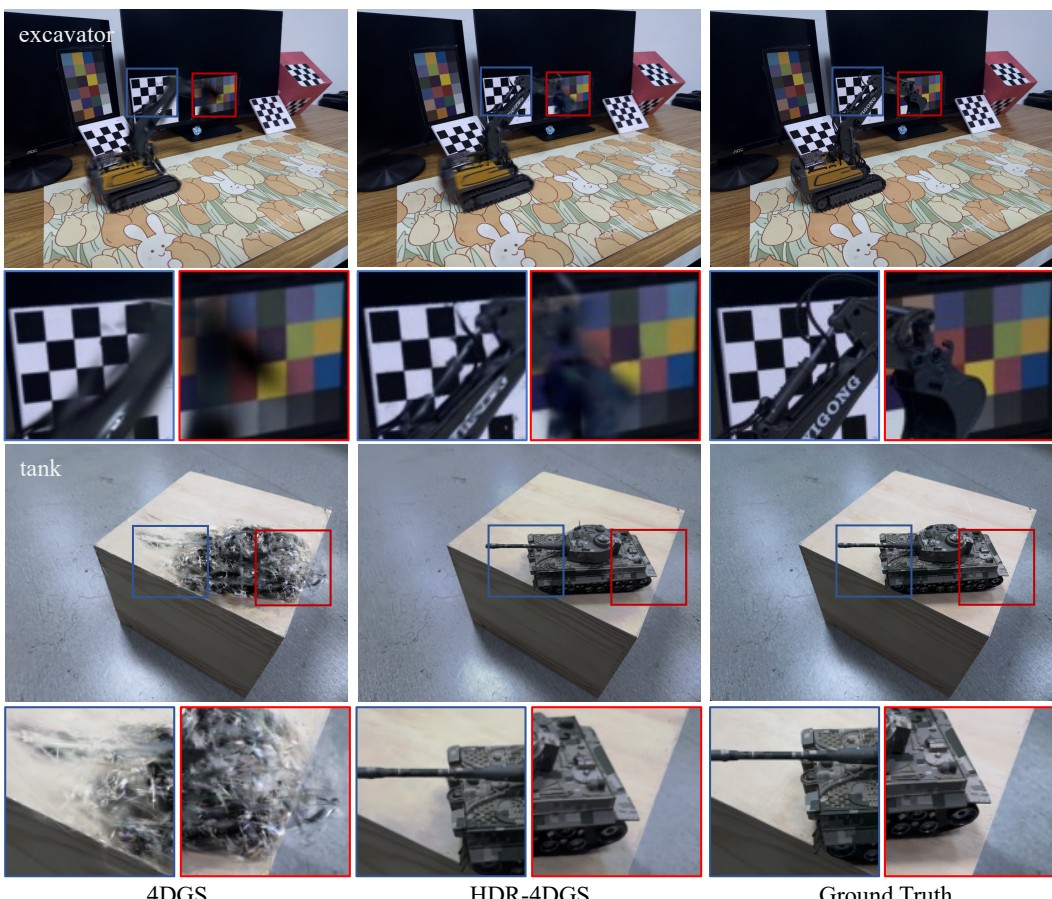

Figure 11: Additional visual comparison of DNVS on HDR-4D-Real.

sults of continuous radiance variation comparisons of HDR DNVS, which again, demonstrate the effectiveness of our method.

## A.3 LIMITATIONS

While effective, our approach currently builds upon the existing 4DGS representation, which was not specifically designed for HDR content. A promising direction for future work is to develop a scene representation explicitly tailored to HDR 4D scenes which incorporats physically grounded priors or adaptive radiance bases to better capture extreme illumination variations and enforce long-range temporal coherence. Another key limitation is the use of a fixed temporal context window in our dynamic tone mapper; an adaptive mechanism that modulates the receptive field based on motion magnitude or radiance variance could significantly improve both computational efficiency

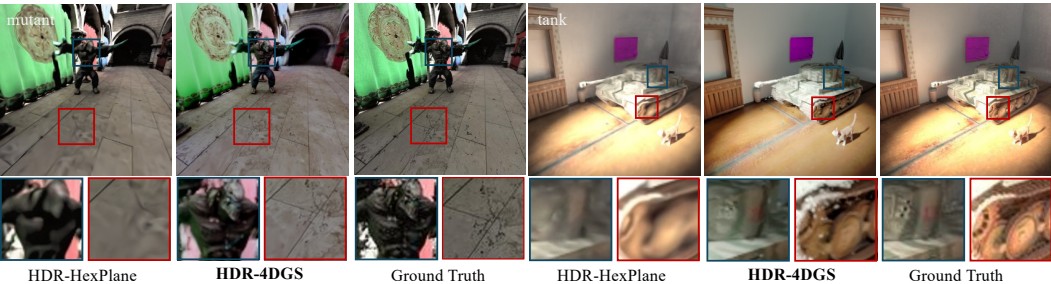

Figure 12: Additional visual comparison of HDR DNVS on HDR-4D-Syn.

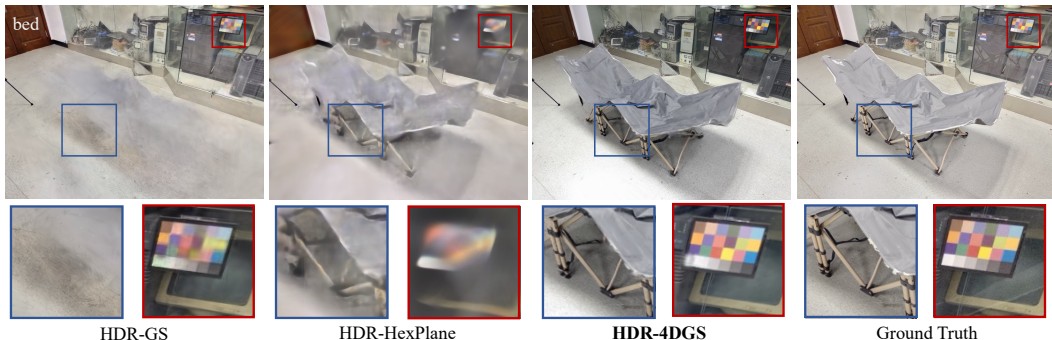

Figure 13: Additional visual comparison of HDR DNVS on HDR-4D-Real.

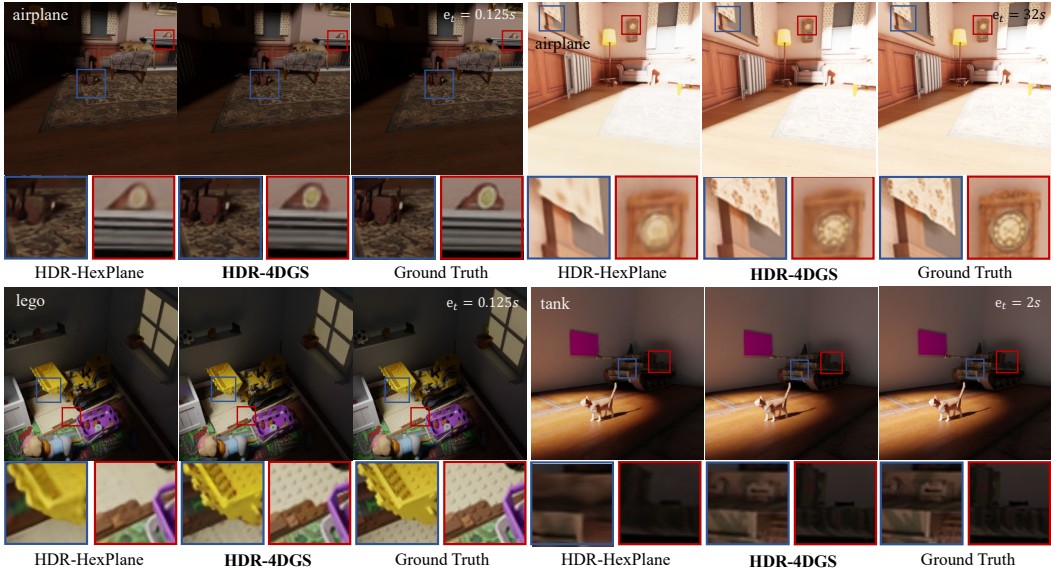

Figure 14: Visual comparison of LDR DNVS on HDR-4D-Syn.

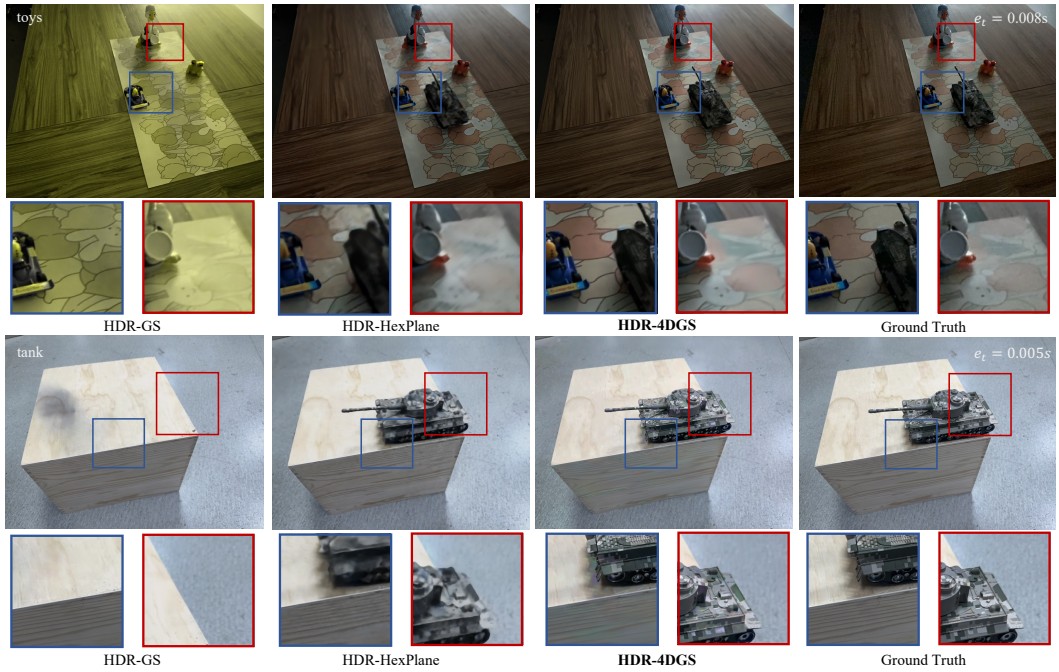

Figure 15: Visual comparison of LDR DNVS on HDR-4D-Real.

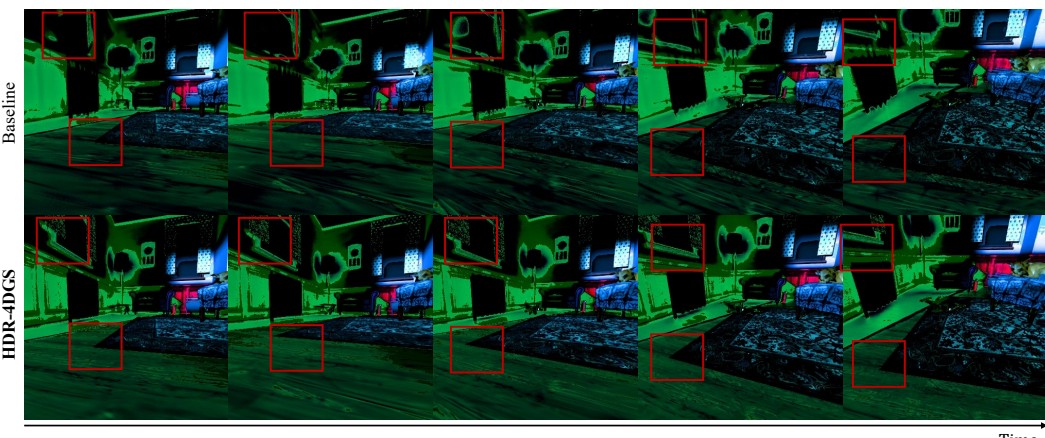

Figure 16: Additional ablation study results of continuous radiance variation comparisons of HDR DNVS on HDR-4D-Syn.

and reconstruction accuracy. Moreover, when foreground and background share similar appearance (*e.g.*, scene jump in HDR-4D-Syn), HDR-4DGS may exhibit suboptimal color reproduction or spatial blurring at dynamic boundaries, suggesting that explicit modeling of semantic or motion boundaries could further disambiguate dynamic content and enhance radiometric consistency across complex scene changes.

## A.4 THE USAGE OF LARGE LANGUAGE MODELS (LLMS)

We clarify that the use of LLMs in this work is strictly limited to polishing the language and presentation of the manuscript. For instance, the original draft description of our datasets in Sec. 4 read:

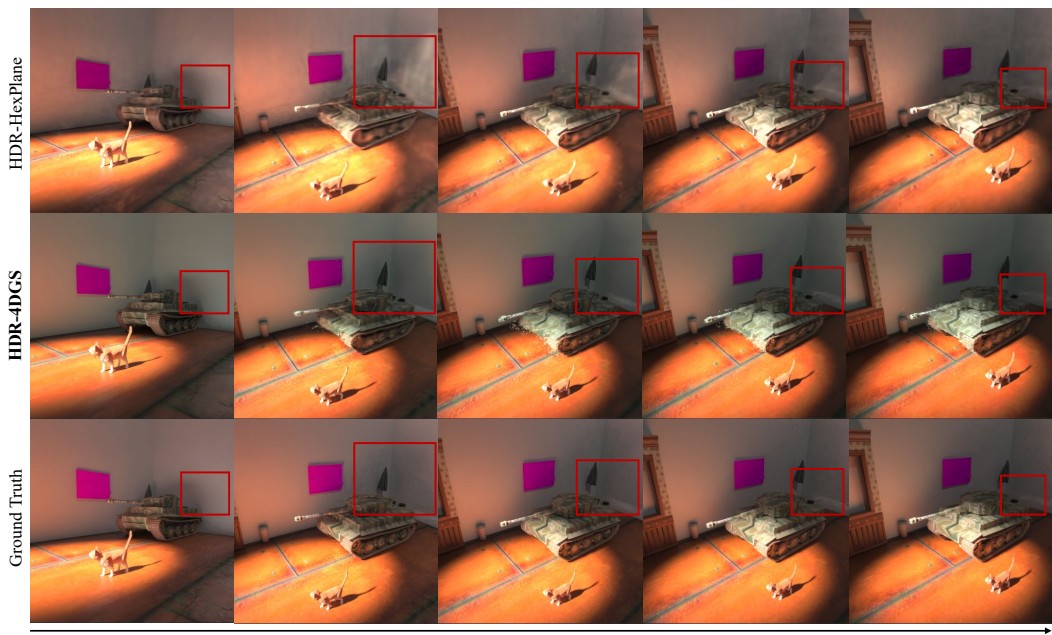

Figure 17: Additional continuous radiance variation comparisons of HDR DNVS on HDR-4D-Syn.

"To address the lack of standardized benchmarks for HDR DNVS, we introduce two novel datasets HDR-4D-Syn and HDR-4D-Real. The synthetic dataset, *HDR-4D-Syn*, comprising 8 synthetic dynamic scenes, is based on the dataset proposed by Wu et al. (2024a) and features videos captured under multi-exposure exposure settings, accompanied by synchronized multi-view LDR video sequences, and corresponding HDR ground truth is re-synthesized. The real-world dataset, *HDR-4D-Real*, consists of 4 real-world indoor dynamic scenes and videos captured under three different exposure times with six iPhone 14 Pro devices, where corresponding HDR images are obtained with UltraFusion (Chen et al., 2025)."

This raw draft was subsequently refined for clarity, grammar, and academic tone — the revised version appearing in the final manuscript reflects only linguistic improvements, with no alteration to technical content or factual claims.

