# OpenReview forum: "Dynamic Novel View Synthesis in High Dynamic Range"
_ICLR.cc/2026/Conference — ICLR 2026 Poster_

### Official Review · Reviewer_yRau · 2025-10-27

**Soundness:** 4
**Presentation:** 3
**Contribution:** 4
**Rating:** 8
**Confidence:** 5

**Summary:**

This paper studies a new problem – high dynamic range dynamic novel view synthesis (HDR DNVS). First of all, the authors collect a 4D synthetic datasets with 8 scenes and a 4D real datasets with 4 scenes. On the set up benchmark, the authors develop a HDR-4DGS, featuring dynamic tone-mapping for adaptively bridging the LDR and HDR domains under complex spatiotemporal variations.

**Strengths:**

(i) This paper studies a brand new problem - high dynamic range dynamic novel view synthesis (HDR DNVS), which has not been explore in the literature. This is novel and interesting. To this end, the authors made a dataset contribution – collecting both synthetic and real datasets to support the research. This is solid, non-trivial, and labor-intensive.

(ii) The proposed biologically inspired dynamic tone-mapping module also shows deep insights. This module draws inspiration from human visual adaptation, where retinal photoreceptors dynamically adjust to ambient brightness. The proposed HDR-4DGS includes a dynamic radiance context learner that models temporal radiance distributions, as shown in Figure 1.

(iii) The writing is good and easy to follow. Especially the method part, all the technical details are very clear. The presentation is also well-dressed. For example, the workflow of the overall pipeline can be clearly shown in Figure 1. In addition, the arrangement of the paper's figures and tables is very neat such as Tables 1, 2 and Figures 2, 3.

(iv) The performance is very solid. As compared in Tables 1 and 2, the proposed HDR-4DGS achieves much better reconstruction quality while keeping a very fast speed. The visual comparisons in figures 2 and 3 also suggest that the proposed HDR-4DGS can render more visually pleasant results with less blur and artifacts.

(v) The ablation study is also very comprehensive. The effects of dynamic tone mapper, dynamic radiance context learner, pixel-level supervision, DRCL design, and the temporal context length are discussed in details.

**Weaknesses:**

(i) Although the authors claim the datasets and the new algorithms are their contributions, there are no datasets or codes submitted. The reproducibility cannot be checked.

(ii) The authors claim the dataset is also a contribution. But there are very few words in the paper introduce how to construct the datasets. It would better if there is a section to describe the details of data collection.

(iii) As the authors claim, the designed method is a 4D Gaussian Splatting. But I did not see any preliminary or method part introduce how to adapt 4DGS to the HDR-DNVS task.

**Questions:**

What is the difference between the designed method with HDR-GS + the k-planes in 4DGS? How about the performance of HDR-GS + the k-planes in 4DGS.

---

> ### Author Response · Authors · 2025-11-21
> **To Reviewer yRau**
>
> > 1. Although the authors claim the datasets and the new algorithms are their contributions, there are no datasets or codes submitted. The reproducibility cannot be checked.
>
> **Re:**  The codes have been submitted in the Supplementary Material. Regarding the datasets (HDR-4D-Syn and HDR-4D-Real), they are 42GB, making direct submission infeasible. We confirm that we will make all datasets publicly available online to ensure full reproducibility.
>
> > 2. The authors claim the dataset is also a contribution. But there are very few words in the paper introduce how to construct the datasets. It would better if there is a section to describe the details of data collection.
>
> **Re:** Great suggestion! As suggested, we have given additional details of data collection in the revised version (see Appendix A.1).
>
> > 3. As the authors claim, the designed method is a 4D Gaussian Splatting. But I did not see any preliminary or method part introduce how to adapt 4DGS to the HDR-DNVS task.
>
> **Re:** Thanks. We provided the preliminary background in Sec. 3.2 on how 4DGS represents dynamic LDR scenes. 4DGS cannot handle the HDR-DNVS task when given multi-exposure LDR images, as it lacks the capability to bridge the HDR and LDR domains. To address that, we adapt its 4DSH to represent the HDR color (see Fig. 1). For the challenge of maintaining temporal radiance consistency, we propose a dynamic tone mapper (DTM), as shown in Fig. 1(b) and Section 3.3. DTM connects the HDR and LDR domains by dynamically adapting per-channel tone-mapping functions in response to evolving temporal radiance patterns.
>
> > 4. What is the difference between the designed method with HDR-GS + the k-planes in 4DGS? How about the performance of HDR-GS + the k-planes in 4DGS.
>
> **Re:** The two methods cannot be directly combined because they utilize fundamentally incompatible scene representations: *explicit primitives* (HDR-GS or 4DGS) versus *implicit feature decoding* (K-Planes).
>
> * *HDR-GS/4DGS:* Using explicit 3D Gaussian primitives, each storing its own parameters such as location, color, opacity.
> * *K-Planes:* Using a feature field encoded by factorized planes. Radiance is inferred implicitly by a separate MLP decoder that maps sampled features to color.
>
> Due to this deep architectural mismatch (fitting parameters to *explicit primitives* vs. inferring values via an *implicit decoder*), combining them is not feasible.

---

> > ### Comment · Reviewer_yRau · 2025-11-21
> >
> > I mean, extending HDR-GS to the temporal axis with the k-plane. What is the difference between the proposed method with the 4D version of HDR-GS?

---

> > > ### Author Response · Authors · 2025-11-22
> > > **Response to Reviewer yRau**
> > >
> > > **Re:** Thanks for the instant engagement and the clarification. The difference between our proposed method and a simple 4D extension of HDR-GS is the inclusion of the Dynamic Tone-Mapping (DTM) module.
> > >
> > > While extending HDR-GS to the temporal axis with k-planes is an interesting idea, it is unclear for us how to integrate these due to their differences as discussed before. We appreciate any specific suggestions and advice on this front.
> > >
> > > Our HDR-4DGS: This is the 4D version of HDR-GS augmented with the DTM. The DTM is our core contribution, specifically designed to model the temporal dynamics in tone mapping, ensuring cross-frame consistency and suppressing flickering in dynamic scenes.
> > >
> > > We did have a 4D version of HDR-GS: A combination of 4DGS (for dynamic geometry) and HDR-GS, compared in our ablation study (see Tab. 4 in revised paper), referred to as "MLP" (part of HDR-GS, used for static tone-mapping). This directly validates the necessity of the DTM module: the static tone mapper achieves a lower PSNR of 23.92, whereas our method with the DTM module achieves significantly better performance at 25.88 PSNR. This quantitative gap confirms that DTM is the critical component for successfully extending HDR-GS to the dynamic domain.
> > >
> > > Hope this clears this question, and please let us know if any more questions. Thanks for the great review efforts and time invovled.

---

### Official Review · Reviewer_u72U · 2025-10-28

**Soundness:** 3
**Presentation:** 2
**Contribution:** 2
**Rating:** 4
**Confidence:** 5

**Summary:**

This paper focuses on learning HDR 3D representations of dynamic scenes from LDR images. Unlike previous methods that are limited to static scenes, this work introduces a dynamic tone-mapping module to achieve adaptive color transformation. By integrating this module with the Gaussian splatting representation, the proposed approach enables HDR rendering from arbitrary viewpoints and time instances. The method is evaluated on both real and synthetic datasets, with some experiments confirming its effectiveness.

**Strengths:**

- The paper extends HDR modeling from static to dynamic scenes by proposing a novel HDR-4DGS framework.
- The paper introduces a dynamic tone-mapping module to handle color translation in dynamic scenes, enabling the rendering of HDR images.
- The method is evaluated on both synthetic and real datasets, and the qualitative and quantitative experiments shown in the paper suggest that it is partially effective in performing color transformations.

**Weaknesses:**

This work proposes a dynamic HDR scene modeling framework that extends Gaussian Splatting to handle dynamic scenes. While the method achieves improvements on part of the experimental data, it still exhibits several concerns, as summarized below：

- Limited performance. As shown in the local results in Figures 2 and 3, the proposed method struggles to recover fine-texture details, and the synthesized novel views exhibit noticeable noise. The authors are encouraged to provide a more detailed analysis of these experimental results to support the effectiveness of the proposed approach better. Furthermore, based on the demo results, the method appears unable to preserve the geometry of moving objects in real-world scenes, often leading to structural loss during deformation. In addition, the DVS results show temporal flickering artifacts, indicating instability in dynamic rendering.

- Limited novelty. According to the paper, the dynamic scene modeling part is adopted from 4DGS, while the per-channel tone-mapping functions adopt the structure from HDR-GS. Thus, the contribution primarily lies in combining these existing components with an additional temporal HDR learning, resulting in an improvement over prior methods that is relatively incremental.

- Insufficient constraints of dynamic Gaussian deformation. As shown in Section 3.4, the method primarily supervises and optimizes the mapping between LDR and HDR domains. Still, it lacks explicit mechanisms or loss functions to control or regularize the deformation of dynamic Gaussians, which are essential for preserving the geometry and spatiotemporal consistency of moving objects. However, such constraints are crucial for dynamic scene representation and novel-view synthesis, and this aspect appears to be missing in this paper.

- Some comparative experiments are missing. Since the proposed method targets dynamic scenes, whereas the comparison methods in the paper mainly focus on demonstrating HDR visual quality, the authors could first map LDR images to HDR and then compare them with dynamic NeRF and Gaussian Splatting approaches to evaluate performance in terms of geometric accuracy and spatiotemporal consistency for moving objects.

**Questions:**

Based on the aforementioned weaknesses, the following issues also require discussion:

- In real-world scenes, as shown in Fig. 2, when using both LDR and HDR images as supervision for HDR rendering, the proposed method performs significantly worse than HDR-HexPlane in quantitative metrics. The authors should provide a clear and thorough analysis to explain this performance gap.

- The paper reports strong results on synthetic data but only limited improvement on real-world datasets. Can you elaborate on whether this discrepancy arises from domain differences, noise in HDR ground truth, or instability in the tone-mapping process?

- The authors could consider providing the output luminance as a function of input intensity for different time frames or varying brightness levels (i.e., a dynamic CRF). This would provide more convincing evidence that the proposed DTM has indeed learned dynamic mapping patterns, rather than relying solely on final HDR rendering metrics.

[Minor]:
- In Figure 1, the notation for the HDR and LDR models appears to be identical, which may cause confusion regarding their respective roles in the framework. And the equations shown in this figure should be formatted consistently with those in the manuscript to ensure clarity and uniform presentation.

---

> ### Author Response · Authors · 2025-11-21
> **Response to Reviewer u72U (Part 1)**
>
> > 1. Limited performance. As shown in the local results in Figures 2 and 3, the proposed method struggles to recover fine-texture details, and the synthesized novel views exhibit noticeable noise. The authors are encouraged to provide a more detailed analysis of these experimental results to support the effectiveness of the proposed approach better. Furthermore, based on the demo results, the method appears unable to preserve the geometry of moving objects in real-world scenes, often leading to structural loss during deformation. In addition, the DNVS results show temporal flickering artifacts, indicating instability in dynamic rendering.
>
> **Re:** We sincerely appreciate the reviewer’s insightful observation. We acknowledge that as a first attempt at HDR Dynamic Novel View Synthesis (HDR DNVS), the problem is inherently ill-posed and "significantly more challenging"  than static settings. However, we argue that our method represents a substantial improvement over existing alternatives with detailed analysis requested below:
>
> **1. Texture and noise:** While some noise is present, we highlight that competitors like HDR-HexPlane suffer from the opposite problem: they tend to produce "overly smooth or blurred reconstructions". As shown in Fig. 2 and Fig. 3, HDR-4DGS recovers intricate structures that competitors miss entirely. We believe a slight trade-off in noise is acceptable for the significant gain in structural detail and HDR fidelity.
>
> **2. Geometry and deformation:** We agree that geometric artifacts exist. These primarily stem from the underlying 4DGS representation, which was not originally designed for HDR content. To provide the "detailed analysis" requested, we conducted an additional experiment where we trained a vanilla 4DGS model on single-exposure LDR imagery (included as Fig. 10 and Fig. 11 in the revised manuscript). The results reveal that standard 4DGS suffers from even *more severe* structural distortion and geometry loss in these scenes compared to our method. This confirms that our HDR modeling strategy effectively mitigates the inherent geometric instability of the base representation.
>
> **3. Temporal flickering:** We respectfully point out that our DTM is explicitly designed to reduce flickering by maintaining "temporal radiance coherence". As visualized in Fig. 5 and analyzed in our ablation study (Tab. 4), our method achieves superior temporal consistency compared to competitors that rely on static tone mappers.
>
> In summary, while limitations remain, including the base representation (we openly discuss in Appendix A.3), HDR-4DGS sets a new state-of-the-art for this novel and difficult task.
>
> > 2. Limited novelty. According to the paper, the dynamic scene modeling part is adopted from 4DGS, while the per-channel tone-mapping functions adopt the structure from HDR-GS. Thus, the contribution primarily lies in combining these existing components with an additional temporal HDR learning, resulting in an improvement over prior methods that is relatively incremental.
>
> **Re:** We respectfully clarify that our contribution goes beyond combining components. We address novelty in two key aspects:
>
> 1. **Problem novelty** (HDR DNVS): We formally introduce this task, which presents "intertwined" geometric and photometric challenges absent in static settings. Existing static methods (*e.g.*, HDR-GS) fail to generalize to this domain, as shown in Tab.s 1 & 2.
>
> 2. **Technical novelty** (DTM): The critical gap in prior work is the lack of dynamics modeling in tone mapping. Our DTM bridges this by extracting a "radiance context embedding" to handle "evolving radiance distributions" across time. This is essential, not incremental: our ablation study confirms that replacing DTM with standard mappers drops performance clearly (Tab. 4: 25.88 vs. 23.92 PSNR).
>
> > 3. Insufficient constraints of dynamic Gaussian deformation. As shown in Section 3.4, the method primarily supervises and optimizes the mapping between LDR and HDR domains. Still, it lacks explicit mechanisms or loss functions to control or regularize the deformation of dynamic Gaussians, which are essential for preserving the geometry and spatiotemporal consistency of moving objects. However, such constraints are crucial for dynamic scene representation and novel-view synthesis, and this aspect appears to be missing in this paper.
>
> **Re:** Great point! We concur that further improving the deformation or dymanics modeling of scene representation would help with model performance. This is orthogonal to our focus of dynamics modeling in tone mapping (see technical novelty above), which we will study in the future. Also note, our method is open to integrate more advanced representation models.

---

> ### Author Response · Authors · 2025-11-21
> **Response to Reviewer u72U (Part 2)**
>
> > 4. Since the proposed method targets dynamic scenes, whereas the comparison methods in the paper mainly focus on demonstrating HDR visual quality, the authors could first map LDR images to HDR and then compare them with dynamic NeRF and GS approaches to evaluate performance in terms of geometric accuracy and spatiotemporal consistency for moving objects.
>
> **Re:** Good point! From the evaluation perspective, model performance has already been assessed by directly comparing the rendered HDR outputs against the HDR ground truth using PSNR, SSIM, and LPIPS. These metrics evaluate HDR fidelity consistently across all methods.
>
> From the training perspective, we also already tested dynamic NeRF/GS models (HexPlane and 4DGS) under HDR supervision (see rows 2 and 4 in Tab. 1 and Tab. 2). As reported, both methods still perform notably worse than our HDR-4DGS, indicating that applying existing models on HDR data does not reliably handle geometric accuracy or spatiotemporal consistency.
>
> These highlight the necessity of a dedicated dynamic HDR reconstruction framework like HDR-4DGS.
>
> > 5. The paper reports strong results on synthetic data but only limited improvement on real-world datasets. In real-world scenes, as shown in Tab. 2, when using both LDR and HDR images as supervision for HDR rendering, the proposed method performs significantly worse than HDR-HexPlane in quantitative metrics. The authors should provide a clear and thorough analysis to explain this performance gap.
>
> **Re:** Great observation! The limited numerical performance on real-world data is attributable to the joint effect of two main factors: *noise in real HDR ground truth* and the *inadequacy of quantitative metrics*.
>
> 1.  *Noise in real HDR ground truth:* Unlike controlled synthetic data, real scenes require fusing multi-exposure LDR inputs into HDR ground truth using methods like UltraFusion (Chen et al., 2025). This 2D-based process introduces multi-view and temporal inconsistencies.
> 2.  *Inadequacy of quantitative metrics:* PSNR fails to fully reflect the holistic quality required for this task (Li et al., 2020). As shown in Fig. 3 and Fig. 9, and also videos (supplementary materials), our method demonstrates **superior perceptual quality** by producing sharper reconstructions and preserving finer details. The quantitative gap is thus a reflection of the metric's failure to accurately score the structural and temporal fidelity gains achieved by our method.
>
> We have discussed this in the revised paper (Sec. 4.1).
>
> > 6. The authors could consider providing the output luminance as a function of input intensity for different time frames or varying brightness levels (i.e., a dynamic CRF). This would provide more convincing evidence that the proposed DTM has indeed learned dynamic mapping patterns, rather than relying solely on final HDR rendering metrics.
>
> **Re:** Great suggestion indeed! We conducted this analysis on representative scenes (*e.g.*, Airplane and Bed) since the motion within them is pronounced enough to affect the scene brightness. After training HDR-4DGS, we extract the learned DTM module along with the radiance bank, both jointly define a *dynamic* mapping from HDR radiance to LDR counterpart, with condition on the time. To assess the dynamics of this learned mapping, we then sample different timestamps and visualize the corresponding tone-mapping curves over the HDR intensity (input).
>
> As illustrated in Fig. 4 (in revised paper), we make these observations with these learned tone mapping curves from HDR radiance to LDR intensities:
> (1) For both scenes, they are all monotonically increasing, consistent with the fundamental definition of a CRF. Also, their curves present distinct patterns, indicating that the learned mapping is scene specific for more dedicated conversion.
>
> (2) For the Airplane scene, the tone mapping curve for the red channel is notably higher than those for the green and blue channels, which is aligned with the scene’s dominant reddish tone, where red intensities significantly exceed those of the other channels. Furthermore, as the airplane transitions from shadowed regions into brighter areas, the overall scene luminance increases, resulting in tone mapping curves that shift upward over time.
>
> (3) Conversely, in the Bed scene, the curves gradually descend as the bed unfolds and the scene becomes progressively darker, reflecting the decreasing overall brightness. These temporal variations in the tone mapping curves demonstrate the model’s ability to adapt dynamically to changing lighting conditions and scene content.
>
> We have added this analysis in Sec. 4.2 of revised paper.

---

> > ### Comment · Reviewer_u72U · 2025-11-26
> >
> > The motivation for introducing the DTM module into the 4DGS framework to achieve HDR rendering is meaningful. However, the limited performance gains still raise concerns about the effectiveness of the proposed method. Moreover, regarding our questions related to the 4DGS aspects, the authors did not provide direct answers and instead repeatedly emphasized improvements on the HDR side. The demo also reveals noticeable shortcomings in both geometric and temporal consistency.

---

> ### Author Response · Authors · 2025-11-21
> **Response to Reviewer u72U (Part 3)**
>
> > 7. In Fig. 1, the notation for the HDR and LDR models appears to be identical, which may cause confusion regarding their respective roles in the framework. And the equations shown in this figure should be formatted consistently with those in the manuscript to ensure clarity and uniform presentation.
>
> **Re:** Thanks! As suggested, we have now used different notations $\mathcal{F}_h$ and $\mathcal{F}_l$ for HDR and LDR, respectively. We have corrected the equations of Fig.1.

---

> ### Author Response · Authors · 2025-11-26
> **Reponse to Revierer u72U**
>
> **Re:** Thanks for instant engagement and new comments. Let us summarize again what we have provided on the three points raised:
>
> - **Limited performance gains**
>
> As the first work and the intrinsic challengs with HDR DNVS, this problem is far from being solved yet. One feature of research is being progressive, as shown in this work. We note that HDR-4DGS achieves **consistent and notable improvements** on both synthetic and real datasets. On the synthetic dataset HDR-4D-Syn, where clean HDR ground truth is available, our performance gains over other competitors are **clear but not limited**, as reflected by established metrics such as PSNR (**+11.18**), SSIM (**+0.216**), and LPIPS (**-0.211**), as shown in Tab. 1.
>
> For the real-world dataset HDR-4D-Real where clean HDR ground truth is not accessible, evaluating HDR rendering quality remains an open problem. Nevertheless, our visual comparisons (*e.g.,* Figs. 2 and 3, Figs. 12 and 13) and the supplementary videos consistently demonstrate **significant perceptual improvements**. We stress again that traditional metrics (*e.g.*, PSNR) may not fully reflect perceptual quality.
>
> In summary, we believe the performance gains of HDR-4DGS provided are **already direct, objective, easily accessible, sufficient rather than limited**, across both synthetic and real-world benchmarks.
>
> We appreciate specific comments on the result gain like what is expected or satisfactory if any.
>
> - **Questions related to 4DGS**
>
> As mentioned in the paper and also our first reponse above, in this work we do not impose new regularizations or loss functions. We agree that having proper design of these components could help, but orthorgonal to our focus on temporal HDR radiance modeling. Here, we do not aim to deliver a comprehensive system with every single aspect covered and solved, as any common research papers. We do not consider this is a valid critique.
>
> - **Shortcomings in the demo**
>
> Again, we stress that this is a new problem, not solved yet by our current method, and more dedicated research is needed to push the progress. Importantly, our proposed method have demonstrated clear superiority in performance over previous state of the art alternatives.
>
> We agree that some of the observed geometric and temporal inconsistencies originate from the base 4DGS representation. To directly address your concern, we added the requested detailed analysis in the revised manuscript (Figs. 10 and 11). These results show that vanilla 4DGS exhibits even *more severe* structural distortion and geometry loss, while our HDR modeling mitigates such instability to a notable extent.
>
> Furthermore, compared with HDR-HexPlane, our proposed HDR-4DGS achieves **better spatial consistency** (Figs. 2, 3, 12, and 13) and **improved temporal coherence** (Figs. 5 and 17).
>
> We acknowledge that HDR DNVS is inherently ill-posed and highly challenging. As the first attempt in this direction, HDR-4DGS may not resolve all issues as is common for emerging research topics. Nevertheless, our method achieves **substantial improvements over existing alternatives** and establishes a **new state of the art** for this novel and difficult task.
>
> We hope that these clarifications address your concerns, and please let us know if any more questions. Thank you again for your valuable review efforts and time.

---

### Official Review · Reviewer_snwX · 2025-10-29

**Soundness:** 3
**Presentation:** 3
**Contribution:** 3
**Rating:** 6
**Confidence:** 3

**Summary:**

This paper addresses the limitations of existing High Dynamic Range Novel View Synthesis (HDR NVS) methods, which are confined to static scenes, and Dynamic Novel View Synthesis (DNVS) approaches that lack HDR support. It formally introduces the task of HDR Dynamic Novel View Synthesis (HDR DNVS) for the first time.

To tackle the core challenges of spatiotemporal radiance consistency and LDR-HDR domain translation in this task, the authors propose the HDR-4DGS framework. Built on 4D Gaussian Splatting, this framework innovatively integrates a Dynamic Tone-Mapping Module (DTM). The DTM stores temporal radiance statistics via a radiance bank, captures temporal variations using a dynamic radiance context learner, and drives per-channel tone-mapping functions to enable adaptive domain translation.

Additionally, the paper constructs two benchmark datasets: HDR-4D-Syn (8 synthetic scenes) and HDR-4D-Real (4 real-world scenes), which provide HDR ground truth and multi-view LDR observations. Experiments demonstrate that HDR-4DGS outperforms existing methods in both quantitative metrics (e.g., PSNR, SSIM) and visual quality, while balancing training efficiency and inference speed, offering an effective solution for high-fidelity HDR novel view synthesis of dynamic scenes.

**Strengths:**

1. The proposed task of HDR Dynamic Novel View Synthesis (HDR DNVS) is a novel and necessary one. Existing methods either focus on static scenes for HDR Novel View Synthesis (HDR NVS) or are limited to Low Dynamic Range (LDR) inputs for Dynamic Novel View Synthesis (DNVS), failing to handle real-world dynamic scenarios with time-varying geometry, illumination, and high-contrast radiance. This gap significantly restricts practical applicability, making the introduction of HDR DNVS a timely and problem-solving initiative.

2. The newly proposed Dynamic Tone-Mapping (DTM) module shows considerable merit, with a clear and logically coherent presentation in the paper. By constructing a radiance bank to store temporal radiance statistics, integrating a dynamic radiance context learner to model temporal variations, and designing per-channel tone-mapping functions, the module explicitly bridges the HDR-LDR domains while maintaining spatiotemporal radiance consistency. Its design draws reasonable inspiration from human visual adaptation, and the technical details (e.g., sliding window for context extraction, joint use of exposure time) are well-explained, making the module’s working mechanism easy to follow.

3. The experimental design is comprehensive and rigorous. First, the authors build two benchmark datasets (HDR-4D-Syn with 8 synthetic scenes and HDR-4D-Real with 4 real-world scenes) that provide essential HDR ground truth and multi-view LDR data for evaluating HDR DNVS, filling the lack of existing benchmarks. Second, they compare HDR-4DGS with multiple state-of-the-art methods (e.g., HDR-NeRF, HDR-GS, HDR-HexPlane) across quantitative metrics (PSNR, SSIM, LPIPS) and qualitative visualizations. Additionally, ablation studies verify the effectiveness of the DTM module and key components (e.g., pixel-level supervision, GRU-based context learner), ensuring the reliability of the results

**Weaknesses:**

1.The scale of the constructed datasets appears relatively limited, which may restrict further in-depth analysis and generalization verification. Specifically, HDR-4D-Syn only includes 8 synthetic scenes and HDR-4D-Real has 4 real-world indoor scenes, with a lack of diversity in scene types (e.g., no outdoor dynamic scenes with complex natural lighting changes, or scenes with more diverse object motions). Such a small dataset not only makes it difficult to fully validate the model’s robustness across different dynamic scenarios but also cannot support additional explorations (e.g., analyzing the model’s performance under varying motion intensities or lighting contrasts). This limitation may weaken the persuasiveness of the model’s practical applicability in broader real-world contexts.

2.The proposed Dynamic Tone-Mapping (DTM) module, though conceptually reasonable, seems relatively general as it lacks testing on the dataset associated with HDR-GS (a representative method for static HDR Novel View Synthesis). HDR-GS is a key baseline in the HDR NVS field, and its dataset likely contains rich HDR-LDR paired data suitable for evaluating tone-mapping performance. Without testing the DTM module on HDR-GS’s dataset, it is hard to fully confirm whether the module can effectively adapt to existing HDR-related data distributions, nor can it more comprehensively demonstrate the module’s advantages over tone-mapping components in static HDR NVS methods. This gap makes the module’s comparative advantage in the broader HDR tone-mapping context less fully validated .

**Questions:**

same as weakness.

---

> ### Author Response · Authors · 2025-11-20
> **Response to Reviewer snwX**
>
> > 1. The scale of the constructed datasets appears relatively limited, which may restrict further in-depth analysis and generalization verification. Specifically, HDR-4D-Syn only includes 8 synthetic scenes and HDR-4D-Real has 4 real-world indoor scenes, with a lack of diversity in scene types (e.g., no outdoor dynamic scenes with complex natural lighting changes, or scenes with more diverse object motions). Such a small dataset not only makes it difficult to fully validate the model’s robustness across different dynamic scenarios but also cannot support additional explorations (e.g., analyzing the model’s performance under varying motion intensities or lighting contrasts). This limitation may weaken the persuasiveness of the model’s practical applicability in broader real-world contexts.
>
> **Re:** This is due to high cost and challenges involved, in particular capturing the real video data.
>
> HDR-4D-Real contains both multi-exposure LDR images and the corresponding HDR ground truth images captured from dynamic scenes. For each scene, data collection was achieved through a labor-intensive and professional process involving multi-camera calibration, pose estimation across varying shutter times, HDR ground-truth synthesis, and strict radiometric consistency checks. In partular, for acquiring the multiple exposure LDR images per camera pose, we have to sweep through all the exposure settings for every pose and every moment -- a very tedious and expensive job. Despite a small number of scenes, they cover representative real-world challenges, including multi-object interactions and occlusions (Toys scene), large-scale rigid motion (Excavator scene), complex non-rigid deformation (Bed scene), and strong, well-defined shadows (Tank scene). All together, these data presents a rich set of challenges for evaluating HDR 4D methods.
>
> While HDR-4D-Syn contains eight scenes, each sequence provides 80–700 multi-view images (1–10 cameras at 800×800 resolution) and spans diverse motion types: rigid motions (Lego, Tank, Deer, Airplane) and fast non-rigid motions (Mutant, Standup, Jump, Hook). Together with synthesized multi-exposure LDR–HDR pairs, the dataset offers a controlled yet challenging benchmark for HDR 4D reconstruction.
>
> In comparsion, this diversity and richness of our datasets are already more than previous 3D/4D datasets such as HDR-Plenoxels [3] (5 synthetic and 4 real static scenes), D-NeRF [4] (8 synthetic dynamic scenes with single exposure) and Neural 3D Video Dataset (Li et al., 2022) (6 real dynamic scenes with single exposure)
>
> While enching data is always preferred (never be too rich), our datasets already provide substantial variation and diversity in motion, illumination, and radiometric conditions to enable meaningful assessment of dynamic HDR reconstruction performance.
>
> [3] Hdr-plenoxels: Self-calibrating high dynamic range radiance fields. ECCV. 2022.
>
> [4] D-nerf: Neural radiance fields for dynamic scenes. CVPR. 2021.
>
> > 2. The proposed Dynamic Tone-Mapping (DTM) module, though conceptually reasonable, seems relatively general as it lacks testing on the dataset associated with HDR-GS (a representative method for static HDR Novel View Synthesis). HDR-GS is a key baseline in the HDR NVS field, and its dataset likely contains rich HDR-LDR paired data suitable for evaluating tone-mapping performance. Without testing the DTM module on HDR-GS’s dataset, it is hard to fully confirm whether the module can effectively adapt to existing HDR-related data distributions, nor can it more comprehensively demonstrate the module’s advantages over tone-mapping components in static HDR NVS methods. This gap makes the module’s comparative advantage in the broader HDR tone-mapping context less fully validated .
>
> **Re:** We thank the reviewer for this suggestion. However, we respectfully clarify that the DTM module is explicitly designed to model temporal radiance variations in dynamic scenes, making the static HDR-GS dataset unsuitable for evaluating its core contribution. Specifically, In the static scenes of the HDR-GS dataset, the radiance distribution remains constant over time. Consequently, the temporal context becomes redundant, and the DTM would theoretically degenerate into a static tone mapper. Evaluating DTM on static data would fail to trigger the module's mechanism for handling evolving radiance, providing no insight into its temporal adaptability.

---

> > ### Comment · Reviewer_snwX · 2025-11-28
> > **Responds**
> >
> > Thanks for authors' clarification, I will keep my score and see about other reviewers' opinions.

---

### Official Review · Reviewer_y3DV · 2025-10-31

**Soundness:** 2
**Presentation:** 2
**Contribution:** 2
**Rating:** 4
**Confidence:** 4

**Summary:**

This paper proposes the High Dynamic Range Dynamic Novel View Synthesis (HDR DNVS) task, aiming to reconstruct spatiotemporally consistent HDR radiance fields and dynamic geometry from sparse, time-varying LDR inputs. It introduces the HDR-4DGS framework based on Gaussian Splatting, featuring a dynamic tone-mapping module (DTM) to bridge HDR-LDR domains while maintaining temporal radiance coherence. Two datasets (HDR-4D-Syn with 8 synthetic scenes and HDR-4D-Real with 4 real indoor scenes) are constructed.

**Strengths:**

1.	The DTM, with a radiance bank and dynamic radiance context learner (DRCL), maintains spatiotemporal consistency, validated by ablation studies .
2.	Constructs dedicated datasets (HDR-4D-Syn/Real) with HDR ground truth and time-varying 3D geometry, enabling standardized evaluation .

**Weaknesses:**

1.	Given that most existing Dynamic Novel View Synthesis (DNVS) methods operate on low dynamic range (LDR) imagery and thus fail under high-contrast or temporally varying lighting, do you consider dynamic HDR modeling essential for achieving photorealistic results in real-world dynamic scenes? In other words, is extending DNVS to HDR space a fundamental requirement to overcome the perceptual and photometric limitations of LDR-based systems, rather than just an optional enhancement?
2.	No comparison with "dynamic scene reconstruction + independent HDR module" combinations, lacking proof of irreplaceability.
3.	When evaluating HDR-4DGS on the HDR-4D-Real dataset, the input LDR images are captured with six synchronized iPhone 14 Pro devices under three exposure times. However, real-world dynamic HDR scenarios may involve more diverse camera models (with different CRFs) or unsynchronized multi-view capture. Why is only a single type of synchronized device used for data collection, and has the method been tested for robustness to camera model differences or capture asynchrony?
4.	In the ablation study on temporal context length (k=20 as optimal), the paper only tests k=5,10,20,30. However, dynamic scenes with varying motion speeds (e.g., fast-moving objects vs. slow illumination changes) may require different k values. Did the authors design experiments to verify whether the optimal k=20 is adaptive to scenes with different dynamic intensities, or is it a fixed hyperparameter that needs manual adjustment for specific scenes?
5.	Is the proposed method able to address issues like extreme lighting performance?

**Questions:**

Please refer to the weakness part.

---

> ### Author Response · Authors · 2025-11-20
> **Response to Reviewer y3DV**
>
> > 1. Given that most existing DNVS methods operate on LDR imagery and thus fail under high-contrast or temporally varying lighting, do you consider dynamic HDR modeling essential for achieving photorealistic results in real-world dynamic scenes?
>
> **Re:** Yes, dynamic HDR modeling is essential, not optional. Just as extending static NVS to DNVS was necessary to handle real-world motion, extending DNVS from LDR to HDR is equally fundamental. Real scenes often exhibit both dynamic content and high-contrast illumination. LDR-based systems cannot adequately capture these characteristics, leading to compromised photorealism. HDR DNVS addresses this fundamental gap in modeling real-world dynamic scenes.
>
> > 2. No comparison to "dynamic scene reconstruction + independent HDR module"
>
> **Re:** Great advice! As suggested, we have done experiments: Training 4DGS on single-exposure (2.0s) LDR images and converting the synthesized LDR images of novel views to HDR images with a variety of existing Single HDR Image Reconstruction models:
>
> | Method                                 | PSNR$\uparrow$ | SSIM$\uparrow$ | LPIPS$\downarrow$ |
> | -------------------------------------- | -------------- | -------------- | ----------------- |
> | 4DGS+EIN (Liu et al., 2025)            | 17.75          | 0.770          | 0.185             |
> | 4DGS+IntrinsicHDR (Dille et al., 2024) | 11.63          | 0.673          | 0.251             |
> | 4DGS+KPNet (Wang et al., 2023)         | 20.92          | 0.813          | 0.197             |
> | 4DGS+KUNet (Wang et al., 2022)         | 19.00          | 0.715          | 0.172             |
> | HDR-4DGS                               | **25.88**      | **0.865**      | **0.076**         |
>
> Results demonstrate that the two-stage pipeline **significantly underperforms** our HDR-4DGS since converting *single-exposure LDR inputs* to HDR is inherently ill-posed where the missing radiance information cannot be reliably recovered in isolation from the scene reconstruction process. Our integrated approach instead jointly optimizes radiance representation and novel-view synthesis, establishing its **necessity and irreplaceability** for high-fidelity HDR scene reconstruction.
>
> We have added this in revised paper (Sec. 4.4).
>
> > 3. Why is only a single type of synchronized device used, test to camera model differences or capture asynchrony?
>
> **Re:** Great question! Using identical synchronized cameras is standard practice in dynamic scene capture, as seen in major datasets such as Neural 3D Video Dataet (Li et al., 2022), Google Immersive Dataset (Broxton et al., 2020), and Technicolor Light-Field Dataset (Sabater et al., 2017). Synchronization ensures temporal consistency crucial for dynamic reconstruction, while identical camera models avoid complications from differing CRFs.
>
> In contrast, asynchronous capture or mixed-camera setups introduce temporal misalignment, inconsistent motion blur, and complex cross-camera radiometric calibration—factors that can obscure the core evaluation of HDR reconstruction. Although improving robustness to such real-world variations is meaningful future work, this study follows established conventions by focusing on synchronized, uniform camera rigs.
>
> We have now added a discussion in Supplement document (see Appendix A.1).
>
> > 4. whether the optimal k=20 is adaptive to scenes with different dynamic intensities, manual adjustment for specific scenes?
>
> **Re:** Great point! We agree that tuning $k$ could further improve performance and will explore this in future work. Note that $k=20$ is not universally optimal—it yields the best *average* result across scenes. Scenes with small motions or slow illumination changes often prefer smaller $k$ (*e.g.*, Mutant performs best at $k=5$), while those with larger motions or more complex dynamics benefit from larger $k$ (*e.g.*, Airplane and Tank peak at $k=20$). This scene dependency aligns with intuition.
>
> Overall, the optimal $k$ is **scene-dependent**, shaped by motion and illumination patterns. Importantly, variations in $k$ do **not drastically affect overall HDR rendering quality** (Tab. 6), indicating the model’s robustness to this parameter.
>
> > 5. Is the proposed method able to address issues like extreme lighting performance?
>
> **Re:** Interesting question! Please note, this work represents the first attempt toward dynamic HDR modeling, but handling extreme lighting conditions, such as very low-light or ultra-high-intensity regions, remains an open challenge often studed in a dedicated sub-field [1,2]. We will extend the findings and experience from this work to further tackling extreme illumination cases in the future, such as further integrating the methods for extreme lighting scenarios.
>
> [1] LL-Gaussian: Low-Light Scene Reconstruction and Enhancement via Gaussian Splatting for Novel View Synthesis. ACM MM. 2025.
>
> [2] A method for optimizing reconstruction quality of 3DGS in complex lighting environments. AOPC. 2025.

---

### Comment · Area_Chair_f927 · 2025-11-27

Dear reviewers,

The authors have submitted their feedback. Please let them know if you have further concerns. Thank you so much!

Best,

Your AC

---

### Meta-Review · Area_Chair_AsHH · 2026-01-05

**Summary:**

The paper proposes HDR-4DGS, which aims to reconstruct spatiotemporally consistent dynamic HDR radiance fields of a dynamic scene from LDR inputs. It received four thorough reviews initially (a rating 8, a rating 6 and two rating 4). The positive reviews acknowledge that 1) the proposed task (HDR DNVS) is a brand new problem, novel and necessary; 2) the proposed dynamic Tone-Mapping is effective; 3) the experiments are comprehensive and rigorous. Concerns are also raised regarding 1) limited technical novelty; 2) lack of additional experimental evaluations etc. The authors provided very detailed rebuttal to the raised concerns/questions. u72U acknowledged the technical contribution of DTM is meaningful after reading the rebuttal in response to the technical novelty concern. After carefully weighing the paper's strengths against its limitations, the AC recommends acceptance of the paper.

**Reviewer Concerns:**

The AC think the raised critical concerns are addressed by the rebuttal.

**Reviewer Scores:**

Based on the raised concerns/weaknesses and the provided rebuttal, the AC think both y3DV and u72U, who give rating 4 initially, might increase their score.

---

### Decision · Program_Chairs · 2026-01-26

Accept (Poster)